



# Greenland ice velocity maps from the PROMICE project

**Anne Solgaard**[1], **Anders Kusk**[2], **John Peter Merryman Boncori**[2], **Jørgen Dall**[2], **Kenneth D. Mankoff**[1],
**Andreas P. Ahlstrøm**[1], **Signe B. Andersen**[1], **Michele Citterio**[1], **Nanna B. Karlsson**[1],
**Kristian K. Kjeldsen**[1], **Niels J. Korsgaard**[1], **Signe H. Larsen**[1], and **Robert S. Fausto**[1]

[1]The Geological Survey of Denmark and Greenland, Østervoldgade 10, 1350 Copenhagen, Denmark
[2]Technical University of Denmark – National Space Institute, Ørsteds Plads 348,
2800 Kongens Lyngby, Denmark

**Correspondence:** Anne Solgaard (aso@geus.dk)

**Abstract.** We present the Programme for Monitoring of the Greenland Ice Sheet (PROMICE) Ice Velocity product (https://doi.org/10.22008/promice/data/sentinel1icevelocity/greenlandicesheet, Solgaard and Kusk, 2021), which is a time series of Greenland Ice Sheet ice velocity mosaics spanning September 2016 through to the present. The product is based on Sentinel-1 synthetic aperture radar data and has a 500 m grid spacing. A new mosaic is available every 12 d and spans two consecutive Sentinel-1 cycles (24 d). The product is made available within ∼ 10 d of the last acquisition and includes all possible 6 and 12 d pairs within the two Sentinel-1A cycles. We describe our operational processing chain from data selection, mosaicking, and error estimation to final outlier removal. The product is validated against in situ GPS measurements. We find that the standard deviation of the difference between satellite- and GPS-derived velocities (and bias) is 20 m yr$^{-1}$ (−3 m yr$^{-1}$) and 27 m yr$^{-1}$ (−2 m yr$^{-1}$) for the components in an eastern and northern direction, respectively. Over stable ground the values are 8 m yr$^{-1}$ (0.1 m yr$^{-1}$) and 12 m yr$^{-1}$ (−0.6 m yr$^{-1}$) in an eastern and northern direction, respectively. This is within the expected values; however, we expect that the GPS measurements carry a considerable part of this uncertainty. We investigate variations in coverage from both a temporal and spatial perspective. The best spatial coverage is achieved in winter due to the comprehensive data coverage by Sentinel-1 and high coherence, while summer mosaics have the lowest coverage due to widespread melt. The southeast Greenland Ice Sheet margin, along with other areas of high accumulation and melt, often has gaps in the ice velocity mosaics. The spatial comprehensiveness and temporal consistency make the product ideal both for monitoring and for studying ice-sheet-wide and glacier-specific ice discharge and dynamics of glaciers on seasonal scales.

## 1 Introduction

The Greenland Ice Sheet (GrIS) is a major contributor to sea-level rise, and approximately half of this contribution is due to ice dynamics (Shepherd et al., 2019). Thus, in order to constrain the ongoing mass loss of the GrIS, it is important to obtain ice-sheet-wide observations of ice-flow velocities. High temporal and spatial resolution will further allow us to distinguish between annual or sub-annual variations and long-term trends, aiding in improving our understanding of the processes behind the observed changes. This is especially important because the flow of glaciers and ice caps varies in

a range of timescales in response to the seasonal cycles, climate change, or internal variability (e.g. Moon et al., 2020; Joughin et al., 2018; Mouginot et al., 2018). In situ measurements of ice-flow velocities (e.g. from GPS) are relatively sparse on the GrIS and are often of short duration (months) at high temporal resolution (e.g. Sole et al., 2011; Maier et al., 2019) or of longer duration (years) but at low temporal resolution (e.g. Thomas et al., 1998; Hvidberg et al., 2020), while few span several years at high temporal resolution (e.g. Ahlstrøm et al., 2013). The sparseness is due to the inaccessibility and size of the GrIS and the harsh climatic conditions, which make fieldwork and instrumentation chal-

lenging. Satellite observations are thus key for deriving time series of ice velocity maps, which can increase our understanding of the dynamics of ice and its interactions with the other components of the climate system.

In the past, surface ice velocity maps of the GrIS and its outlet glaciers only resolved annual or seasonal characteristics due to the limited availability of data (e.g. Rignot and Kanagaratnam, 2006; Howat et al., 2010; Moon et al., 2014; Joughin et al., 2010). This scenario changed with the launch of Landsat 8 (a joint NASA–USGS mission) in 2013 and ESA's Sentinel-1 (2014 and 2016) and Sentinel-2 (2015 and 2017) satellites. With this, freely available data became abundant, especially increasing the temporal resolution and spatial coverage of ice velocity products. At present, several freely available GrIS-wide velocity products exist with different temporal resolution, e.g. annual, quarterly, and monthly mosaics from Copernicus Climate Change Service, ESA Climate Change Initiative (CCI) (Nagler et al., 2015), and NASA's MEaSUREs programme (e.g. Joughin, 2020a, c, b) including the ITS_LIVE project (Gardner et al., 2019) based on synthetic aperture radar (SAR) and/or optical data. These are updated periodically with a lag. Scene pair velocities over Greenland from Landsat are available from NASA MEaSUREs ITS_LIVE at present up until 2018 (Gardner et al., 2019) and from TU Dresden covering the period 1972–2015 (Rosenau et al., 2015). Furthermore, Mouginot et al. (2019a, b, c, d) provide a freely available product comprising annual ice velocity mosaics for the GrIS spanning the period 1972 to 2017 based on both optical and SAR data.

Here we present the Programme for the Monitoring of the Greenland Ice Sheet (PROMICE) (http://www.promice.org, last access: 1 July 2021) Ice Velocity product, which is a time series of ice velocity mosaics based on Sentinel-1 SAR offset tracking. This work is part of the PROMICE monitoring effort focusing on the GrIS. The product benefits from the abundance, continuity, and high temporal resolution of the Sentinel-1 SAR data and is continuously updated every 12 d. The product is used as input to, for example, the solid ice discharge product by Mankoff et al. (2020) on a routine basis and to study GrIS-wide glacier dynamics in high temporal detail in Vijay et al. (2019). In the following sections we describe the ice velocity product, the data it is derived from, the operational setup, and the data processing steps that are used to generate it. Finally, we make use of available GPS measurements in order to validate our velocity product.

## 2   The PROMICE Ice Velocity product

The PROMICE Ice Velocity product (https://doi.org/10.22008/promice/data/sentinel1icevelocity/greenlandicesheet, Solgaard and Kusk, 2021) is a geospatial time series of Greenland-wide ice velocity mosaics produced using the Interferometric Post Processing (IPP) processor (see Sect. 4).

The product spans the period 13 September 2016 to present and has a grid spacing of 500 m and a temporal resolution of 24 d. The effective spatial resolution is of the order of 800–900 m, determined by the fixed size of the correlation windows used in the offset tracking (see Sect. 4.1). Thus, glaciers smaller than approximately 1 km across will not be fully resolved. The product is based on measurements of displacements between pairs of radar images acquired 6 or 12 d apart (see Sect. 3.1). To achieve a consistent coverage (see Sect. 6), each mosaic is based on velocity measurements from all possible 6 and 12 d pairs using data from Sentinel-1A and Sentinel-1B within two consecutive Sentinel-1A orbit cycles (i.e. 24 d). A new map is produced for every Sentinel-1A cycle, i.e. a new mosaic every 12 d. A given mosaic thus overlaps by 12 d with the previous and subsequent maps. The dataset is expanded continuously, and we aim to provide a new mosaic within 10 d of the last acquisition. However, during the winter campaigns where more data are acquired, this lag may be larger. The velocity provided at every grid point in the PROMICE Ice Velocity product is the weighted average of all velocity measurements available at that grid point within that 24 d period (see Sect. 4.3) and should be considered an average estimate of velocity over the 24 d period during which the radar images were acquired (see further discussion in Sect. 6). The start and end times of this period are given in the *time_bnds* variable in the PROMICE NetCDF product (see below). Figure 1 shows samples of the time series at different times during the year 2020.

Each ice velocity mosaic is supplied as a single NetCDF file following the Climate and Forecast (CF) conventions (see https://cfconventions.org/, last access: 1 July 2021). The mosaics are provided on a 500 m polar stereographic Greenland-wide grid with latitude of true scale at 70° N and reference longitude −45° E (EPSG 3413 projection). The variables in the NetCDF product are listed in Table 1. A quick-look image for each mosaic is provided along with the dataset.

## 3   Data

In the following, we present the characteristics of the Sentinel-1 data and introduce the input data that we use to generate the PROMICE Ice Velocity product.

### 3.1   Sentinel-1 SAR data characteristics

SAR sensors are well suited for polar observations because data collection is not impacted by the polar night or cloud cover. The Sentinel-1 constellation currently consists of two satellites, Sentinel-1A and Sentinel-1B, equipped with identical C-band (5.4 GHz) SAR sensors. Over the GrIS, the Sentinel-1 SAR mainly employs the interferometric wide swath (IW) mode (De Zan and Guarnieri, 2006) allowing for generation of radar images with a resolution of approximately 3 m on the ground in the slant range (line-of-sight di-

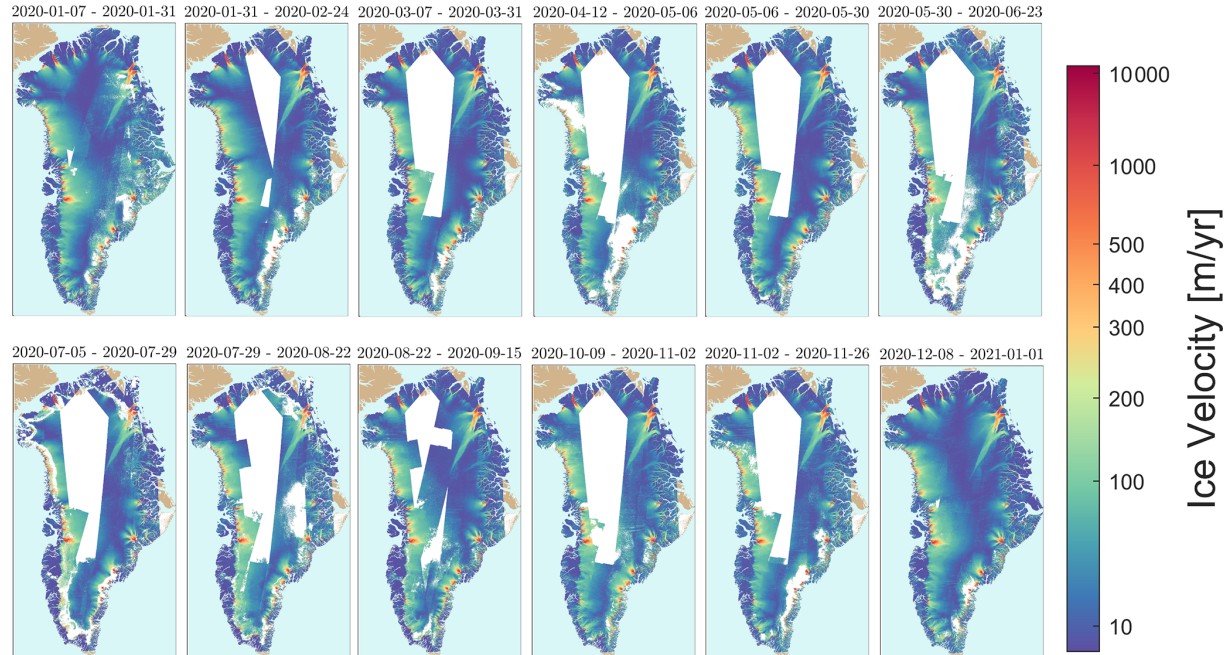

**Figure 1.** Examples of the PROMICE Ice Velocity maps from 2020: from top left corner to lower right approximately one map per month over 2020.

**Table 1.** Variables in the PROMICE Ice Velocity NetCDF product.

| Variable | Description | Unit |
|---|---|---|
| $x$ | $x$ coordinate of projection | m |
| $y$ | $y$ coordinate of projection | m |
| time | Midpoint time of all contributing acquisitions | Days since 1 January 1990 |
| time_bnds | First and last time of contributing acquisitions | Days since 1 January 1990 |
| land_ice_surface_easting_velocity | Ice velocity along $x$ axis | $\mathrm{m\,d^{-1}}$ |
| land_ice_surface_northing_velocity | Ice velocity along $y$ axis | $\mathrm{m\,d^{-1}}$ |
| land_ice_surface_vertical_velocity | Vertical velocity from surface parallel flow | $\mathrm{m\,d^{-1}}$ |
| land_ice_surface_velocity_magnitude | Horizontal ice velocity magnitude | $\mathrm{m\,d^{-1}}$ |
| land_ice_surface_easting_velocity_std | Ice velocity error estimate along $x$ axis | $\mathrm{m\,d^{-1}}$ |
| land_ice_surface_northing_velocity_std | Ice velocity error estimate along $y$ axis | $\mathrm{m\,d^{-1}}$ |
| land_ice_surface_velocity_magnitude_std | Horizontal ice velocity error estimate | $\mathrm{m\,d^{-1}}$ |

rection) and 22 m in the azimuth (flight-path direction). The pixel spacing of the product is 2.3 m in slant range and 14.1 m in azimuth.

The near-polar orbit has a repeat cycle of 175 orbits, corresponding to 12 d, with the two satellite orbits phased 6 d apart. With the current observation schedule, the entire margin of Greenland is imaged every 12 d by both satellites (Fig. 2a). Furthermore, the entire ice sheet is mapped from several additional tracks every winter from December to February, allowing the generation of Greenland-wide maps during this season (Fig. 2b).

SAR-based ice velocity measurements are based on processing of image pairs. Images acquired within short time intervals (temporal baselines) retain a high degree of coher-

ence (see Sect. 5.3) and therefore measurement coverage; however they also exhibit an increased sensitivity to error sources, which do not depend on the temporal baseline (e.g. processing artefacts, ionospheric and orbit estimation errors; see Sect. 5).

Image pairs can be formed between acquisitions from the same satellite, i.e. S1A–S1A or S1B–S1B, with temporal baselines which are a multiple of 12 d, i.e. 12, 24 d, etc. In addition, image pairs can also be formed from acquisitions obtained from two different satellites, i.e. S1A–S1B or S1B–S1A, with a temporal baseline which is an odd multiple of 6 d, i.e. 6, 18 d, etc. Although in principle the radar instruments on board the Sentinel-1A and Sentinel-1B satellites are identical, there are some subtle differences, which should

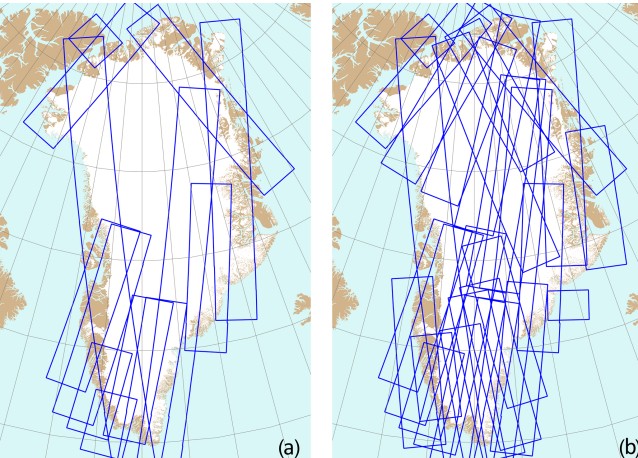

**Figure 2.** Typical Sentinel-1 coverage over Greenland for a single 12 d orbital cycle **(a)** during the standard observation scenario and **(b)** during the dedicated winter campaign from December 2019–February 2020. The blue polygons represent acquisitions from different tracks, acquired at different times during the cycle.

be taken into account when selecting the data pairs for processing, as detailed in Sect. 5.2.

## 3.2 Input data

The data used for generating the PROMICE Ice Velocity product are single-look complex (SLC) IW radar images (with annotation), supplied by the Copernicus Open Access Hub (https://scihub.copernicus.eu/, last access: 1 July 2021). SLC images are focused SAR images referenced to the radar acquisition geometry and have the highest resolution of the available IW product types (3 m × 22 m). They are supplied as slices with a footprint of approximately 250 km × 250 km. Owing to the peculiarities of the IW mode, each slice is subdivided into three range swaths, named IW1–IW3, which are acquired in an interleaved (bursted) fashion; these swaths are stored in separate files. The SLC images are supplemented by restituted orbit files, available a few hours after acquisition, and precise orbit files, available 21 d after the data acquisition at https://scihub.copernicus.eu/gnss/ (last access: 1 July 2021). Since the PROMICE Ice Velocity product is typically generated before the latter become available, we use the restituted orbit files as we have found that the difference between the restituted orbit files and the precise orbit files is insignificant for our data products (see Sect. 5.1).

In order to geocode measurements made in radar geometry, a digital elevation model (DEM) is used. We employ the Greenland Ice Mapping Project (GIMP) DEM based on the ASTER and SPOT 5 DEMs and AVHRR photoclinometry (Howat et al., 2014, 2015), downsampled to 500 m spacing to match the resolution of the ice velocity product.

## 4 Methods

The data processing is carried out using the IPP processor, developed and maintained by DTU Space (Kusk et al., 2018). Despite the name, the processor also performs offset tracking for displacement measurements, which is the functionality used to generate the PROMICE product. It is a highly automated processing chain requiring little user intervention. The processing is described in the following section, and a high-level overview of the processing flow is shown in Fig. 3a.

To support the PROMICE product generation, a database with all available SLC products over Greenland is maintained. This is updated daily by searching the Copernicus Open Access Hub. An automated system downloads all new SLC data to a central storage location. Product generation is initiated by an operator selecting a Sentinel-1A reference orbit cycle number. All SLCs from Sentinel-1A in that and the following cycle (24 d of data) are first selected for processing. Additionally, all SLCs from Sentinel-1B acquired within the same 24 d time span are selected for processing. Then, all possible SLC pairings with a 6 or 12 d baseline are calculated, and the offset-tracking processing described in Sect. 4.1 is automatically carried out for each pair. When all pairs required for a product have been processed, the geocoding and error estimation described in Sects. 4.2 and 5 is performed for each pair, followed by fusion and mosaicking of all the pairs, as described in Sect. 4.3.

## 4.1 Offset tracking

The offset-tracking procedure employed is similar to the one described in Strozzi et al. (2002) and estimates local shifts between two SLCs in radar geometry using normalized cross-correlation of intensity image patches. It is illustrated in Fig. 3b.

The SLC with the earliest acquisition time is used as the reference image. Prior to the processing, calibration constants in range and azimuth timing are applied to correct for the different geolocation biases observed in Sentinel-1A and Sentinel-1B SLCs (see also Sect. 5.2). For 12 d pairs, these constants are identical for both SLCs and have no effect.

An output grid is defined in the reference image geometry, with a spacing of 40 pixels in the slant-range direction and 10 pixels in the along-track (azimuth) direction. These spacings correspond to approximately 150 m × 150 m on the ground.

For intensity cross-correlation methods, the SLCs need not be coregistered and resampled to sub-pixel accuracy prior to the processing, as they rely on relatively large windows (several tens of pixels in each dimension). Instead, for the regular grid points in the reference SLC, we calculate the expected position of the corresponding grid points (assuming no motion) in the second SLC, based solely on SLC timing information, orbital state vectors, and the DEM. The grid points are selected on integer pixel positions in the reference SLC, but the corresponding grid points will generally not coincide

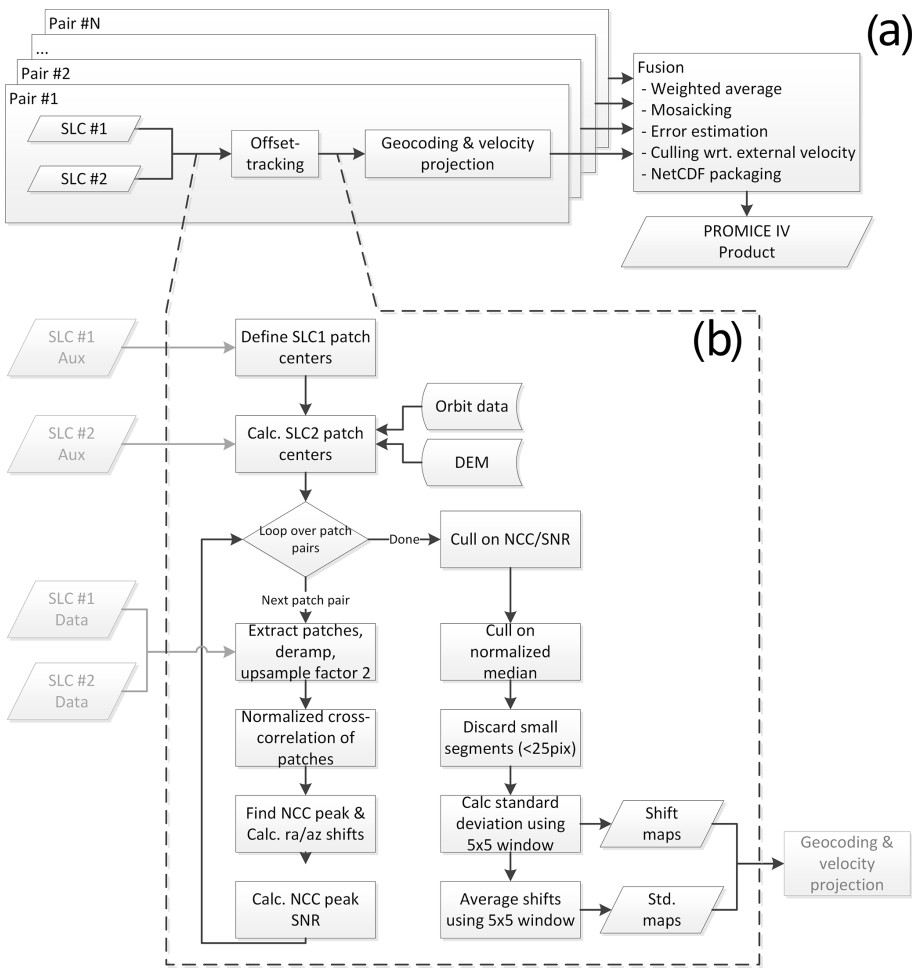

**Figure 3.** Processing flow for PROMICE Ice Velocity product: **(a)** high-level processing flow and **(b)** offset-tracking details.

with integer pixel locations in the second SLC. To avoid re-sampling the second SLC, we round the corresponding grid points to their nearest integer pixel locations and save the fractional shifts, which are then added back to the offset measurement after cross-correlation.

At each grid point, surrounding image patches of $256 \times 64$ complex pixels (slant range × azimuth) are extracted in both SLCs. This patch size has been chosen to maximize the coverage over different flow regimes and coherence levels, and it means that the product has an effective spatial resolution of the order of 800–900 m. As shown in Merryman Boncori et al. (2018), an adaptive window size approach similar to the one described in Joughin (2002) could provide for a locally finer spatial resolution when the data allow it, but this is currently not implemented in the IPP processor. Each patch is deramped (Miranda, 2017) and upsampled by a factor of 2 (in both range and azimuth) using fast Fourier transform (FFT) interpolation, and the intensity (magnitude squared of the complex pixel values) is derived. A normalized cross-correlation of the two upsampled real-valued patches is carried out, resulting in a correlation surface with values be-

tween 0 and 1. The integer shift between the two patches is then estimated by locating the peak of the correlation surface, and a $9 \times 9$ neighbourhood surrounding the peak is upsampled by a factor of 4, again using FFT interpolation. Then the fractional shift is retrieved by fitting a parabola to the peak and its two surrounding pixels in each dimension, correcting finally for oversampling factors and accounting for the fractional shift initially estimated for the second SLC. A signal-to-noise ratio (SNR) for the peak estimate is calculated by dividing the correlation value of the peak by the mean of the surrounding pixels in the correlation surface (de Lange et al., 2007). The estimated 2-D shift, the peak normalized cross-correlation value (NCC), and the SNR are all saved for further processing.

The procedure described above will yield a shift estimate even if the two images are completely uncorrelated, so a culling of the estimated shifts is carried out. First, pixels with an NCC < 0.05 or SNR < 7 are set as invalid. Then, a further culling is carried out on the range and azimuth shifts, using a normalized median test, as described in Westerweel and Scarano (2005). For each measurement, $U_0$, in a

$5 \times 5$ neighbourhood, the median, $U_m$, of the 24 surrounding measurements $(U_1, U_2, \ldots, U_{25})$ is calculated (excluding $U_0$), and for each measurement in the neighbourhood a residual, $R_i = |U_i - U_m|$, is calculated. The median, $R_m$, of $(R_1, R_2, \ldots, R_{24})$ is then calculated and used to normalize the residual of $U_0$ so that $R_0' = |U_0 - U_m|/(R_m + \epsilon)$, where $\epsilon$ is a minimum normalization level that accounts for cross-correlation noise. We use $\epsilon = 0.1$ pixels, as suggested in Westerweel and Scarano (2005), and cull the measurement, $U_0$, if $R_0'$ exceeds a threshold of 5 for either of the range or azimuth shifts. This value has been found by experiments to remove most clearly visible outliers, without removing valid measurements. Lower values remove more outliers but have an adverse effect on measurement coverage. After the culling, small unconnected segments of pixels ($< 25$ pixels) are removed, as these were found to often contain erroneous values. Some outliers may remain, especially in areas subject to surface melt, as the associated strong radar backscatter can create false correlation peaks. An additional culling based on time series statistics (see Sect. 4.3) is carried out on the final mosaicked product to further suppress these outliers.

To aid in error estimation, the local standard deviations of the two shift maps (range and azimuth shifts) are estimated in a sliding $5 \times 5$ window, ignoring pixels with invalid measurements, and the shift maps are finally averaged by a $5 \times 5$ window. The shift maps (in units of SLC pixels) and associated standard deviations are stored along with the SLC parameters and orbit information to be used in the subsequent processing.

## 4.2   Geocoding and horizontal velocity projection

The geocoding takes as input the shift maps and associated standard deviation maps output by the offset tracking and the DEM. Using the DEM and orbit information, the maps are interpolated to the output grid in map projection (see Sect. 2). The shifts and standard deviations are converted to velocity by multiplying with the SLC pixel spacing and dividing by the temporal baseline. At this stage, the velocities and standard deviations, even though provided on a georeferenced grid, are still measured in the radar range/azimuth geometry. With a single pair providing only two velocity measurements, it is not directly possible to estimate three-dimensional flow. Instead, we assume surface parallel flow (SPF) and estimate the flow as described in the following. Let $\boldsymbol{v}_{xyz} = [v_x, v_y, v_z]^T$ be the three-dimensional velocity vector in map geometry and $\boldsymbol{v}_{SAR} = [v_r, v_a]^T$ be the velocity vector in radar geometry, with $v_r$ the range (line-of-sight) velocity and $v_a$ the azimuth (along-track) velocity. With the SPF assumption, the vertical velocity component becomes (Joughin et al., 1998)

$$v_z = \left( \frac{\partial z}{\partial x} v_x + \frac{\partial z}{\partial y} v_y \right), \qquad (1)$$

where $\left( \frac{\partial z}{\partial x}, \frac{\partial z}{\partial y} \right)$ is the surface gradient derived from the DEM. The effective resolution of the velocity maps is of the order of the correlation window size, which corresponds to approximately $800 \times 900$ m on the ground (see Sect. 4.1), so the resolution of the surface gradient map should approximately match this. The DEM is downsampled to the pixel spacing of the PROMICE product ($500 \times 500$ m), and the gradient is derived using second-order differences, which means the gradients are derived using samples approximately 1000 m apart. The relation between the horizontal and the radar velocity can be written as

$$\begin{bmatrix} v_r \\ v_a \end{bmatrix} = \begin{bmatrix} \cos\theta\cos\phi + \sin\theta\frac{\partial z}{\partial x} & \cos\theta\sin\phi + \sin\theta\frac{\partial z}{\partial y} \\ -\sin\phi & \cos\phi \end{bmatrix} \begin{bmatrix} v_x \\ v_y \end{bmatrix}, \qquad (2)$$

where angles $\phi$ and $\theta$ describe the orientation of the line-of-sight (LoS) vector pointing from the pixel under consideration to the sensor, with the horizontal angle $\phi$ measured anticlockwise from the $y$ axis of the map projection and the elevation angle $\theta$ measured from the local horizontal plane to the LoS vector. The horizontal velocity components (and the associated standard deviation maps) can then be found by inversion of Eq. (2). Projection scaling factors are not applied to the velocities, so these represent physical velocities along the projection axes.

## 4.3   Fusion

The fusion step describes the process of combining and mosaicking the geocoded offset-tracking results onto a Greenland-wide grid. For every pixel on the output grid, we do a weighted averaging of the $N$ valid velocity measurements from all pairs covering the pixel, using as weights the inverse of the measurement variances:

$$\hat{v} = \sum_{n=1}^{N} \frac{1}{\sigma_n^2} v_n \cdot \left( \sum_{n=1}^{N} \frac{1}{\sigma_n^2} \right)^{-1}, \qquad (3)$$

where $\hat{v}$ is the fused ($x$ or $y$) velocity, $v_n$ is the ($x$ or $y$) velocity measurement from pair $n$, and $\sigma_n$ is the associated standard deviation. The estimated standard deviation of the pixel is then

$$\hat{\sigma} = \sqrt{\left( \sum_{n=1}^{N} \frac{1}{\sigma_n^2} \right)^{-1}}. \qquad (4)$$

## 4.4   Culling

After all measurements have been fused and mosaicked, temporal culling is carried out to remove further outliers. This relies on comparison of the measured value with an average value of all available measurements, based at the time

of writing on more than 4 years of data. For each pixel, we reject the measurement if

$$\frac{\sqrt{\left(\hat{v}_x - v_{m,x}\right)^2 + \left(\hat{v}_y - v_{m,y}\right)^2}}{\sqrt{v_{m,x}^2 + v_{m,y}^2} + v_\epsilon} > k_{thr}, \tag{5}$$

where $(\hat{v}_x, \hat{v}_y)$ is the fused velocity measurement; $(v_{m,x}, v_{m,y})$ is the average velocity; $v_\epsilon$ is a velocity constant preventing erroneous culling in areas with very low velocities; and $k_{thr}$ is a constant factor, setting the threshold for culling. A low value of $k_{thr}$ will remove more outliers but may also remove valid measurements in areas with strong seasonal variation, such as glaciers with significant speed-up during the melt season. This effect is showcased for ice velocity along the flow line from Hagen Bræ in north Greenland (Fig. 4a). The slow-flowing outlet glacier experiences periods of speed-up during summer, where velocities near the terminus increases more than 200 %. At the same time surface melt inhibits processing parts of the data, resulting in spikes in the ice velocity as evident in Fig. 4b. Figure 4c and d show how values of $k_{thr} = 3$ and 1 cull the data. For $k_{thr} = 3$, the (real) summer speed-up near the front is conserved, while the majority of spikes further inland are removed. Applying a stricter value, $k_{thr} = 1$, removes not only outliers but also the real signal due to summer speed-up. In this case, 8 % of the pixels are culled, while 4 % of the pixels are culled when applying $k_{thr} = 3$. For the PROMICE Ice Velocity product we apply $v_\epsilon = 20\,\text{m yr}^{-1}$, whereas $k_{thr} = 3$. This choice of threshold is a balance of removing as much noise as possible without removing actual signal in the mosaics encompassing a wide range of ice dynamics.

Figure 5 provides an example of the effect of applying $k_{thr} = 3$ to a map from summer 2018 when surface melt influences the data quality. The unculled and culled maps are displayed in Fig. 5a and b. The location of the culled data points is shown in red in Fig. 5c. Note for example the removal of the noisy areas in the west and east Greenland ablation zone (Fig. 5c, d, and e) and locations influenced by ionospheric stripes in the slow-moving interior. On average, $\sim 2\,\%$ of the pixels in a mosaic are culled using this procedure. More pixels are culled in summer mosaics than in winter mosaics (see Fig. 10a).

## 5   Error sources and estimation

In this section, we describe the error sources affecting the PROMICE Ice Velocity product in more detail. The error sources can be divided into five main groups:

1. slowly varying errors, such as those caused by orbit errors (Sect. 5.1) or other timing biases in the products (Sect. 5.2);

2. temporal decorrelation caused by changes in the radar backscatter between observations (see Sect. 5.3);

3. ionospheric errors resulting in localized but spatially correlated errors in the measured azimuth shift (see Sect. 5.4);

4. aliasing errors caused by the need to acquire two observations from which we infer displacement and then velocity (any extreme velocity highs or lows will be smoothed out);

5. errors due to tidal motion of floating glacier tongues.

### 5.1   Orbit errors

Errors in the Sentinel-1 orbital state vectors provided by ESA will result in an apparent shift between the two SLCs in a pair, translating directly into biases in the velocity measurement. For Sentinel-1 data, absolute orbital errors are of the order of 5 cm rms when using the precise orbit product available after 21 d (Peter et al., 2017). The restituted orbits typically used in the PROMICE product generation are available shortly after acquisition and have a nominal accuracy of 10 cm rms. This corresponds to $8.6\,\text{m yr}^{-1}$ rms for a measurement using a 6 d pair and $4.3\,\text{m yr}^{-1}$ for a 12 d pair. To assess the difference between using restituted and precise orbits, we processed 18 different Sentinel-1 12 d pairs (9 S1A–S1A and 9 S1B–S1B pairs) acquired consecutively over an area in southwest Greenland where much of the scene in the IW1 swath consists of bedrock. The reason for using only 12 d pairs is to exclude the effect of S1A–S1B biases, which are treated instead in Sect. 5.2. All pairs were processed twice, using either the precise orbits or the restituted orbits, with all other parameters identical. The processing carried out consisted of offset tracking and geocoding (see Sect. 4.1 and 4.2). Averaging for each processed pair the measured range and azimuth velocities over the bedrock area, which can be assumed stationary, gives an estimated average residual velocity error for that pair. The mean of the 18 residual range and azimuth velocity estimates and associated standard deviations are listed in Table 2. We note that the range velocity bias ($-0.5\,\text{m yr}^{-1}$) and azimuth velocity bias ($0.5\,\text{m yr}^{-1}$) do not differ between the precise and the restituted orbit files. The standard deviation in the range direction is $2.7\,\text{m yr}^{-1}$ for the precise orbit files and $2.8\,\text{m yr}^{-1}$ for the restituted orbit files, while the standard deviation in the azimuth direction is $3.3\,\text{m yr}^{-1}$ for both orbit types. Overall, the error statistics for the two orbit types are almost completely identical, and the use of restituted versus precise orbits has an insignificant impact on the accuracy of the final velocity products.

### 5.2   Geolocation bias correction

With the commissioning of Sentinel-1B in late 2016 it became possible to generate ice velocity products with a 6 d

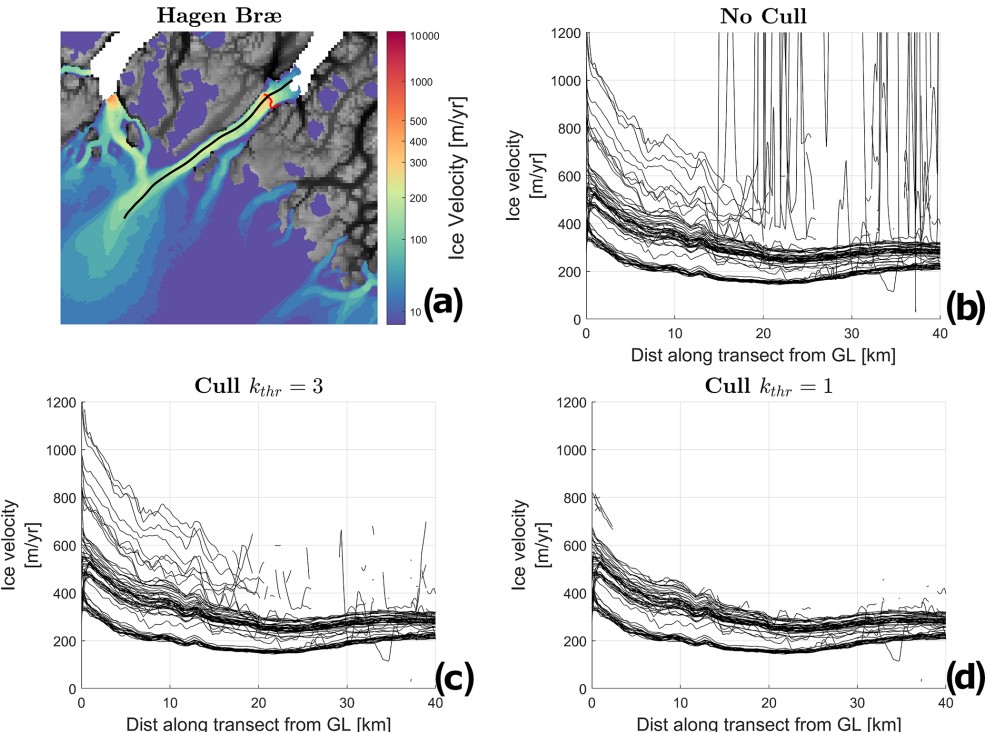

**Figure 4.** The effect of the culling procedure: example from Hagen Bræ, north Greenland. **(a)** Overview of the Hagen Bræ area. The flow line is plotted in black, and the grounding line (GL) (from ESA Greenland Ice Sheet CCI) is plotted in red. The ice velocity along the flow line in panel **(a)** is plotted for all maps since September 2016 in the case of no culling **(b)**, applying $k_{thr} = 3$ **(c)**, and applying $k_{thr} = 1$ **(d)**.

**Table 2.** Comparison of precise and restituted orbit files. Range direction is line of sight, and azimuth direction is along the satellite flight path.

| Orbit type | Range velocity bias (m yr$^{-1}$) | Range velocity SD (m yr$^{-1}$) | Azimuth velocity bias (m yr$^{-1}$) | Azimuth velocity SD (m yr$^{-1}$) |
|---|---|---|---|---|
| Precise | −0.5 | 2.7 | 0.4 | 3.3 |
| Restituted | −0.5 | 2.8 | 0.4 | 3.3 |

temporal baseline by combining Sentinel-1A and Sentinel-1B data (see also Sect. 3.2). Although the SAR instruments are in theory identical, our analysis of the initially generated 6 d ice velocity products revealed velocity biases not present in 12 d (same satellite) pairs. To quantify this, an experiment was carried out using 37 Sentinel-1 pairs (9 6 d AB pairs, 10 6 d BA pairs, and 18 12 d AA and BB pairs grouped together). The 18 12 d pairs are identical to the pairs used in Sect. 5.1; thus, all pairs were acquired over an area in southwest Greenland where much of the scene in the IW1 swath consists of bedrock. Averaging the measured range and azimuth velocity over the bedrock area gives an estimated average residual velocity error for that pair. Figure 6a shows a scatter plot of the residual range velocity ($V_r$) versus residual azimuth velocity ($V_a$) for the 37 pairs. The corresponding mean and standard deviation values are listed in Table 3.

As expected, the 12 d statistics are identical to those observed in the orbit type comparison (see Sect. 5.1 and Table 2), with a bias magnitude below 0.5 m yr$^{-1}$. For the 6 d pairs (middle part of Table 3), the bias magnitudes are significantly larger and change sign, depending on whether the first SLC in the pair is acquired from Sentinel-1A or Sentinel-1B. The standard deviations of the 6 d bias estimates are approximately 2 times those of the 12 d estimates, which is expected, as the velocity measurements are based on measurements of shifts between the images, which are then divided by the temporal baseline to arrive at velocities. The average bias magnitudes for the 6 d pairs are 8.8 m yr$^{-1}$ in range and 28.8 m yr$^{-1}$ in azimuth. With the 6 d baseline, this corresponds to bias magnitudes on the measured shifts of 0.15 m in range and 0.48 m in azimuth. These values are consistent with results obtained in detailed analysis of Sentinel-1 SLC product geolocation using corner reflectors; see Schubert et al. (2017)

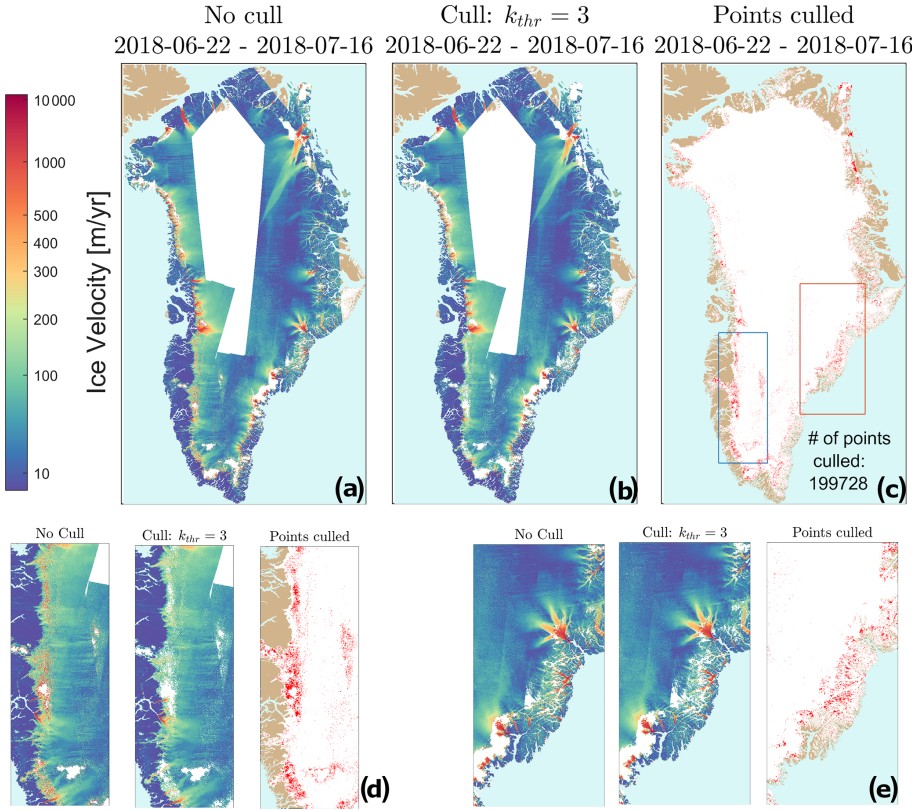

**Figure 5.** The effect of the culling procedure on the GrIS scale for a map from summer 2018: **(a)** the ice velocity map with no culling applied. **(b)** The same ice velocity map with a culling threshold of $k_\text{thr} = 3$ applied. **(c)** The locations of the culled points are shown as red dots. **(d)** Zoom-in on the blue box in western Greenland. **(e)** Zoom-in on the red box in eastern Greenland.

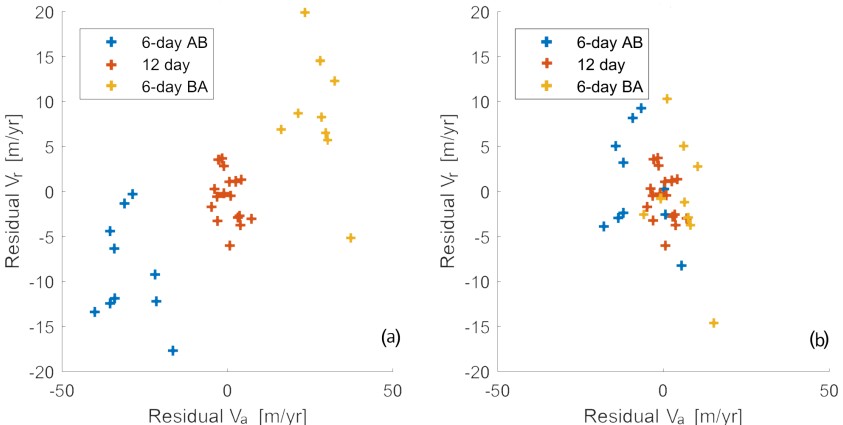

**Figure 6.** Scatter plot of IW1 average residual range ($v_r$) and azimuth ($v_a$) velocity error of 37 Sentinel-1 pairs: **(a)** without calibration and **(b)** calibrated with constants from Gisinger et al. (2020).

and Gisinger et al. (2020). The latter reports average shifts between Sentinel-1A and Sentinel-1B of 0.16 m in range and 0.40 m in azimuth, corresponding to velocities of 9.7 and 24.4 m yr$^{-1}$, respectively, for measurements using 6 d pairs, but it also suggests that there may be a swath dependence of these shifts. Our analysis above concerns only the IW1

swath, so for now we use the constants from Gisinger et al. (2020) mentioned above to calibrate the PROMICE product. The calibration is implemented as an adjustment to the timing annotation for the SLC products prior to the offset tracking. Applying these calibration constants to the test dataset described above results in a significantly reduced bias in the

**Table 3.** Velocity biases for different satellite combinations.

| Pair type | Range bias (m yr$^{-1}$) | Range SD (m yr$^{-1}$) | Azimuth bias (m yr$^{-1}$) | Azimuth SD (m yr$^{-1}$) |
|---|---|---|---|---|
| 12 d AA and BB | −0.5 | 2.7 | 0.4 | 3.3 |
| Uncalibrated 6 d BA | −8.9 | 5.7 | −29.9 | 7.6 |
| Uncalibrated 6 d AB | 8.6 | 6.9 | 27.5 | 6.3 |
| Calibrated 6 d BA | 0.6 | 5.7 | −7.9 | 7.6 |
| Calibrated 6 d AB | −0.9 | 6.9 | 5.5 | 6.3 |

6 d measurements, as shown in Fig. 6b and in the bottom part of Table 3. The calibrated 6 d range velocities now have a mean bias magnitude of 0.8 m yr$^{-1}$, and the azimuth velocities a mean bias of 6.7 m yr$^{-1}$. In the final PROMICE Ice Velocity product, the weighted averaging of 12 d pairs and both AB and BA 6 d pairs will tend to reduce the impact of any residual biases (see also Sect. 4.3).

## 5.3  Temporal decorrelation

Temporal decorrelation is caused by changes in radar backscatter between acquisitions that reduce the correlation between the image patches which are cross-correlated in the offset-tracking procedure, leading to noisy or even missing measurements. The surface of the interior of the ice sheet is relatively homogeneous, with no large-scale features, and the velocity measurement relies on preservation of the speckle pattern (coherence) between observations (Gray et al., 1998). Speckle is a property of radar images, caused by variations in the sub-resolution structure of the imaged scene, resulting in large pixel-to-pixel intensity fluctuations in otherwise homogeneous areas. If the ice flow is spatially uniform and the sensor track does not deviate excessively for the two acquisitions (the latter is generally not a problem for Sentinel-1), the speckle pattern can be tracked between acquisitions using the cross-correlation procedure described in Sect. 4.1. Precipitation, surface melt, and steep spatial gradients in ice flow can all reduce the coherence and thus the ability to measure ice velocity in such areas. Often in the interior, the signal-to-noise ratio is low, but since five velocity maps from each track are averaged to produce the PROMICE product, the noise can be reduced. In extended homogeneous areas of low coherence, the velocity measurements can become patchy, since many unreliable measurements will be discarded by the culling procedures described in Sect. 4.1 and 4.4.

On outlet glaciers, the rapid ice flow and associated deformation tends to destroy the coherence except for short temporal baselines. Here, the ability to measure displacement relies instead on the presence of larger-scale features, such as crevasses, which can be still be tracked between images with the cross-correlation procedure described in Sect. 4.1, even if there is no coherence.

Models that express the shift errors as a function of coherence do exist (De Zan, 2014), but the coherence cannot directly be used to estimate errors or discard measurements, since velocity measurements can often still be made in non-coherent, fast-moving areas, as mentioned above. In the PROMICE Ice Velocity product, the velocity error estimate is based instead on the local standard deviation of the tracked shifts (see Sects. 4.1 and 5.6).

## 5.4  Ionospheric errors

Ionospheric propagation errors arise due to spatial fluctuations (scintillations) in the ionosphere total electron content within the SAR synthetic aperture length (i.e. kilometre-scale variations) (Gray et al., 2000). This is especially a problem in the near-polar regions. For a given image pixel, these fluctuations cause an azimuth variation in the raw signal phase, which is not accounted for by the SAR focusing, resulting in an azimuth shift of the focused pixel. The varying propagation naturally also causes a shift in the range direction, but these shifts are much smaller (typically on a centimetre scale) than those observed in the azimuth direction (comparable to the azimuth pixel size, i.e. several metres; Mattar and Gray, 2002). The shifts vary along the scene according to the ionosphere conditions along the satellite flight path, often present in only parts of the scene. Also the observed shifts are strongly correlated in the range direction, appearing as linear or slightly curved "streaks" superposed on the azimuth shift map. In the PROMICE velocity mosaics, such streaks are readily identifiable by the human eye, appearing as roughly east–west-oriented stripes of varying intensity. The velocity errors caused by ionosphere can exceed 200 m yr$^{-1}$, impacting 6 d pairs twice as much as 12 d pairs, since the shifts caused by the ionosphere do not depend on the temporal baseline. An example of the impact of the ionosphere can be seen in Fig. 7, showing a single-pair 6 d velocity measurement and a 12 d velocity measurement, the former exhibiting significant ionospheric streaks. Both pairs share a common SLC (acquired on 11 October 2016), suggesting in this case that the ionosphere effects can be attributed to the other SLC of the 6 d pair.

Methods for reducing the impact of the ionosphere on ice velocity measurements typically rely on the dispersive na-

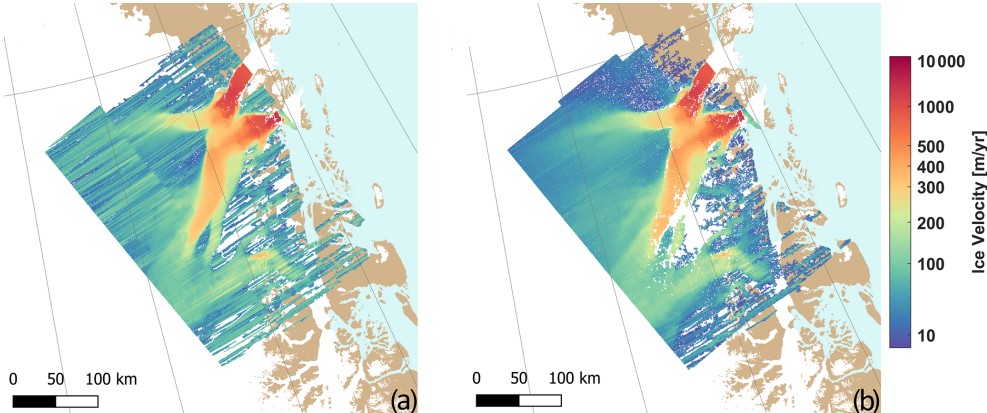

**Figure 7.** Two single-pair velocity maps from relative orbit 74 (ascending), illustrating the impact of ionospheric streaks: **(a)** 6 d pair acquired on 11 and 17 October 2016, with strong ionosphere errors, and **(b)** 12 d pair acquired on 11 and 23 October 2016, with limited ionosphere errors.

ture of the ionosphere delay and have been applied to L-band interferometric ice velocity measurements (Liao et al., 2018). A method for correcting azimuth shift measurements in Sentinel-1 data has been proposed in Gomba (2018) but has not been demonstrated for Sentinel-1 ice velocity measurements. Another method for reducing the impact of ionospheric effects is to exploit the fact that in some regions measurements from both ascending and descending tracks are available, and in this case ice velocities can be derived from only the range offsets – which are much less sensitive to ionospheric effects – and the SPF assumption (see Sect. 4.2). In the standard S1 acquisition scenario (Fig. 2a), only two ascending tracks are acquired (the long track along the west coast of Greenland and the track covering the northeast margin of the ice sheet up to the northernmost point), so the method is not always applicable everywhere. During the winter campaigns (Fig. 2b), this method will be applicable over a much larger part of the ice sheet. Work is underway to include this method in a future update of the PROMICE product.

The mitigation of ionospheric effects in the PROMICE Ice Velocity product relies on culling and averaging. Pixels with large ionospheric errors, if present in regions with generally low velocities, will be removed by the temporal culling procedure described in Sect. 4.3. In areas where multiple velocity observations are available, the weighted averaging in the fusion (see Sect. 4.3) will tend to reduce, but not completely remove, the ionospheric effects. We estimate that the ionospheric effects can cause a velocity error of up to 300 m yr$^{-1}$ and will mainly affect the $v_y$ component, which is roughly aligned with the azimuth direction due to the near-polar orbit of the Sentinel-1 satellites.

### 5.5 Tidal motion

A few outlet glaciers in Greenland (Petermann and 79 Fjord being the most prominent) are characterized by having a floating tongue subject to tidal motion. The tidal motion introduces a vertical shift, with the sensitivity of this shift to the tidal signal increasing from 0 near the grounding line to 1 on the fully floating part of the tongue, a transition zone which is typically 5–10 km wide (Padman et al., 2018). This vertical shift will affect the ice velocity estimate, as the difference in the shift between the two radar acquisitions is projected onto the radar line of sight and interpreted as motion in the slant range direction. In Reeh et al. (2000), tidal-induced shifts of approximately ±0.5 m were observed using GPS receivers placed on the floating part of 79 Fjord glacier. This could, for a 6 d pair, lead to errors in the ice velocity estimate of more than 50 m yr$^{-1}$, although the averaging of several acquisitions in the PROMICE product will tend to reduce this error. The effect is not modelled in the PROMICE product, so care should be taken when using the product on floating glaciers.

### 5.6 Error estimation

The error estimates provided with the PROMICE Ice Velocity product are derived from the local standard deviation of the underlying shift maps generated by the offset tracking (see Sect. 4.1 and 4.3). As such, they do not account for slowly varying errors, such as those described in Sect. 5.1 and 5.2, and account only to a limited extent for the impact of ionospheric errors, as these are locally correlated on the scale of the window size used to estimate the local standard deviations. Although this is not a complete error characterization, it was shown in Merryman Boncori et al. (2018) to provide the correct order of magnitude for the errors. Examples of relative error estimates accompanying the two ice velocity maps from Fig. 7 are shown in Fig. 8. The strong

ionospheric streaks evident in the 6 d pair in Fig. 7 are seen to be reflected in the corresponding error estimate, although the magnitude is underestimated. In the central and lower left part of the maps, errors are seen to be generally higher in the 12 d pair, but in a more diffuse pattern, even though the 12 d pair is less sensitive to a given shift error, due to the longer baseline. In this case, it is the higher temporal decorrelation of the 12 d pair that causes an increased noise level, which is also reflected in the error estimate. A Greenland-wide view of the error estimate for the PROMICE product is given in Fig. 9 for the same mosaics displayed in Fig. 1.

The slowly varying errors (Sect. 5.1 and 5.2) could potentially be corrected by calibrating the measured velocities using ground control points (GCPs), either on stable terrain or in areas where the ice flow is known to vary little. In practice this is difficult to do in an automated system, as the calibration has to be carried out on the individual pairs, where the ionospheric and, to some extent, the temporal decorrelation errors associated with offset tracking are often much larger than the slowly varying errors. If GCPs are unwittingly selected in areas affected by, for instance, the ionosphere, then the GCP calibration could actually have a detrimental impact. A large number of GCPs, well distributed in the image, would be required to reduce the statistical noise, but this can often not be achieved within the limited spatial coverage of a single pair. For this reason, the PROMICE Ice Velocity product is not calibrated using GCPs.

# 6   Properties of the PROMICE Ice Velocity product

The PROMICE Ice Velocity product is designed as a compromise between good spatial coverage, high temporal resolution, and low noise. Other combinations of 6 and 12 d pairs are possible resulting in a different temporal resolution and spatial coverage. We explore other possibilities for products and compare them to the PROMICE Ice Velocity product with respect to coverage and noise. These products are time series of mosaics consisting of (see Table 4)

1. all 6 d pairs (no 12 d pairs) within two Sentinel-1A cycles (**6dOnly**),

2. all 12 d pairs (no 6 d pairs) within two Sentinel-1A cycles (**12dOnly**),

3. all 6 d pairs (no 12 d pairs) within one Sentinel-1A cycle (**6dOnly_1cycle**).

Time series number 3, 6dOnly_1cycle, thus has twice the frequency compared to the PROMICE product.

Data coverage for each mosaic in each time series is displayed in Fig. 10a. We defined the coverage as the fraction of grid points that contain ice velocity data on the ice sheet in a given mosaic. We have included a time series in the analysis called **All-pairsNoCull**, which includes the same data as the PROMICE product but has not undergone the culling procedure described in Sect. 4.4. The percentage of points that are culled in each mosaic of the PROMICE product using that method is also shown in Fig. 10a. If all grid points on the ice sheet contain data, then coverage is 1. All time series have close to full coverage during peak winter, where a campaign ensures full IW coverage of the ice sheet over a number of cycles. The coverage of the PROMICE product drops to $\sim 0.7$ outside the campaigns, with a low during summer months. Figure 10b shows the mean of the reported standard deviation for each mosaic in the time series. In general, when coverage is low the noise goes up, and vice versa. 6dOnly_1cycle is the time series with the highest reported error estimate.

Figure 11 provides a spatial view of the coverage of the time series. It shows the percentage of all mosaics that have data in a given grid point for each of the time series. Blue colours indicate that a grid point rarely has data throughout the time series, while yellow indicates a temporal coverage close to 100 %. A number of circumstances influence the temporal coverage: the more acquisitions cover a grid point, the more likely it is to have a pair where coherence is not lost. The number of acquisitions depends on the time of year, the location, and the time span of the product (the PROMICE product includes more pairs than any of the three other time series). The temporal coverage also depends on how often coherence is lost, leading to how often the processing fails.

All four time series have a large low-coverage area in the ice-sheet interior, where SAR data in IW mode are rarely acquired as is evident from Fig. 2. The same explanation is true for the smaller triangular areas in the Melville Bay area and northern Greenland as well as the Scoresbysund area (locations 1, 2, and 3 in Fig. 11a). However, the large area with low coverage along the southeast ice sheet margin, an area in southern Greenland, one north of Rink Glacier in west Greenland, and one in the Melville Bay area all have routine SAR IW acquisitions every 6 d (locations 4, 5, 6, and 7 in Fig. 11a). This will be discussed in the following.

The coverage and quality of each mosaic depend on the SAR data coverage, on the number of acquisitions going into each mosaic, and on how the properties of the ice-sheet surface have changed between acquisitions (Sects. 5 and 4.3 and Fig. 2a). The PROMICE product has the best coverage of the time series in Table 4 (excluding All-pairsNoCull) (Figs. 10 and 11) as it includes all the pairs contained in both 12dOnly and 6dOnly. Figure 10a shows that most often 6dOnly has better coverage than 12dOnly, and for some extended periods of time it is comparable to the PROMICE product. However, both 12dOnly and 6dOnly time series have periods with significant drops in coverage, while the PROMICE product still performs well. The difference in coverage is caused by differences in data acquisition and ice-sheet surface properties.

Not all tracks have both 12 d and 6 d coverage, and often tracks in the interior are only covered by 12d pairs. This is revealed by the lower coverage of the interior for 12dOnly compared to 6dOnly in Fig. 11. 6dOnly has better coverage along the ice sheet margins compared to 12dOnly, because coherence is more likely to be preserved for shorter tempo-

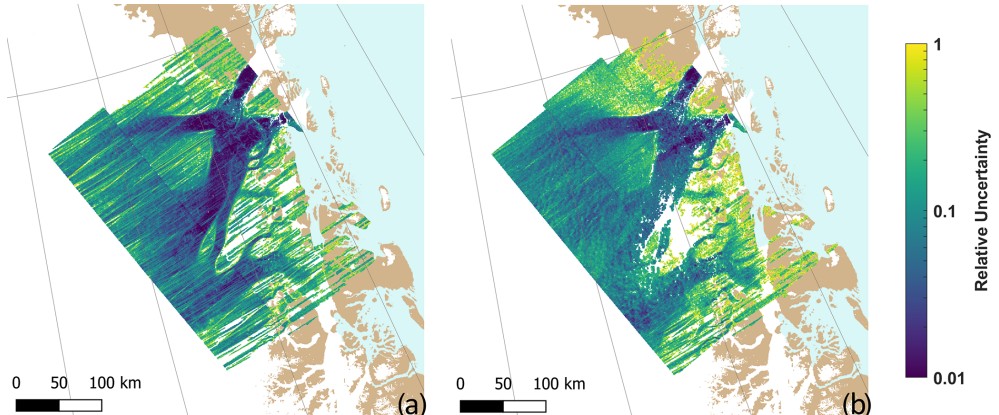

**Figure 8.** Relative horizontal velocity error estimates for the single-pair velocity maps in Fig. 7: **(a)** 6 d pair acquired on 11 and 17 October 2016 and **(b)** 12 d pair acquired on 11 and 23 October 2016.

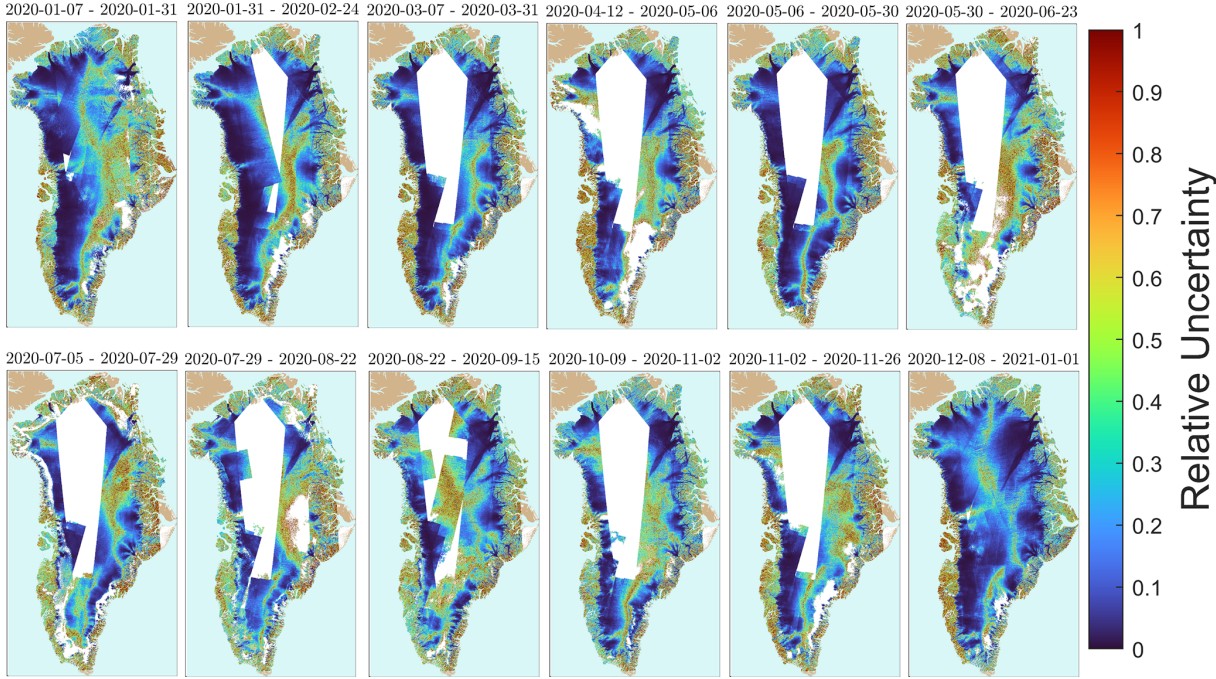

**Figure 9.** Greenland-wide view of the error estimates for the PROMICE mosaics displayed in Fig. 1.

ral baselines as discussed earlier in Sect. 3.1. The PROMICE product mosaics thus have better coverage than both 6dOnly and 12dOnly, because they each have coverage where the other does not. However, even when the short 6 d temporal baseline is used, some areas consistently have low coherence and therefore rarely have ice velocity coverage. The largest of these areas is the southeast ice sheet margin, while the smaller areas include an area in southern Greenland, one north of Rink Glacier in west Greenland, and one in the Melville Bay area (Fig. 11). The areas are apparent in all time series but are most pronounced in the 12dOnly series as a longer temporal baseline increases the probability of changes to the surface properties due to precipitation and/or

surface melt. The areas discussed here largely coincide with the regions identified as high accumulation percolation areas (HAPAs) by Vandecrux et al. (2019), studying firn properties. HAPAs are areas on the ice sheet characterized by frequent precipitation events and surface meltwater that percolates into the firn – both processes leading to loss of coherence.

Figure 10 also shows the effect of performing the culling described in Sect. 4.4 on the time series as a whole. AllpairsNoCull is the PROMICE product without culling. The effect on the coverage is minor (Fig. 10a); however Fig. 10b shows that the average noise level is significantly reduced.

https://doi.org/10.5194/essd-13-1-2021 Earth Syst. Sci. Data, 13, 1–22, 2021

**Table 4.** Information on time series of mosaics from Sentinel-1 data. All-pairsNoCull contains the same data as the PROMICE product but has not undergone the culling procedure described in Sect. 4.4.

|  | Temporal resolution | All 12 d pairs included | All 6 d pairs included |
| --- | --- | --- | --- |
| PROMICE product (all pairs) | 24 d | × | × |
| 6dOnly | 24 d |  | × |
| 12dOnly | 24 d | × |  |
| 6dOnly_1cycle | 12 d |  | × |
| All-pairsNoCull | 24 d | × | × |

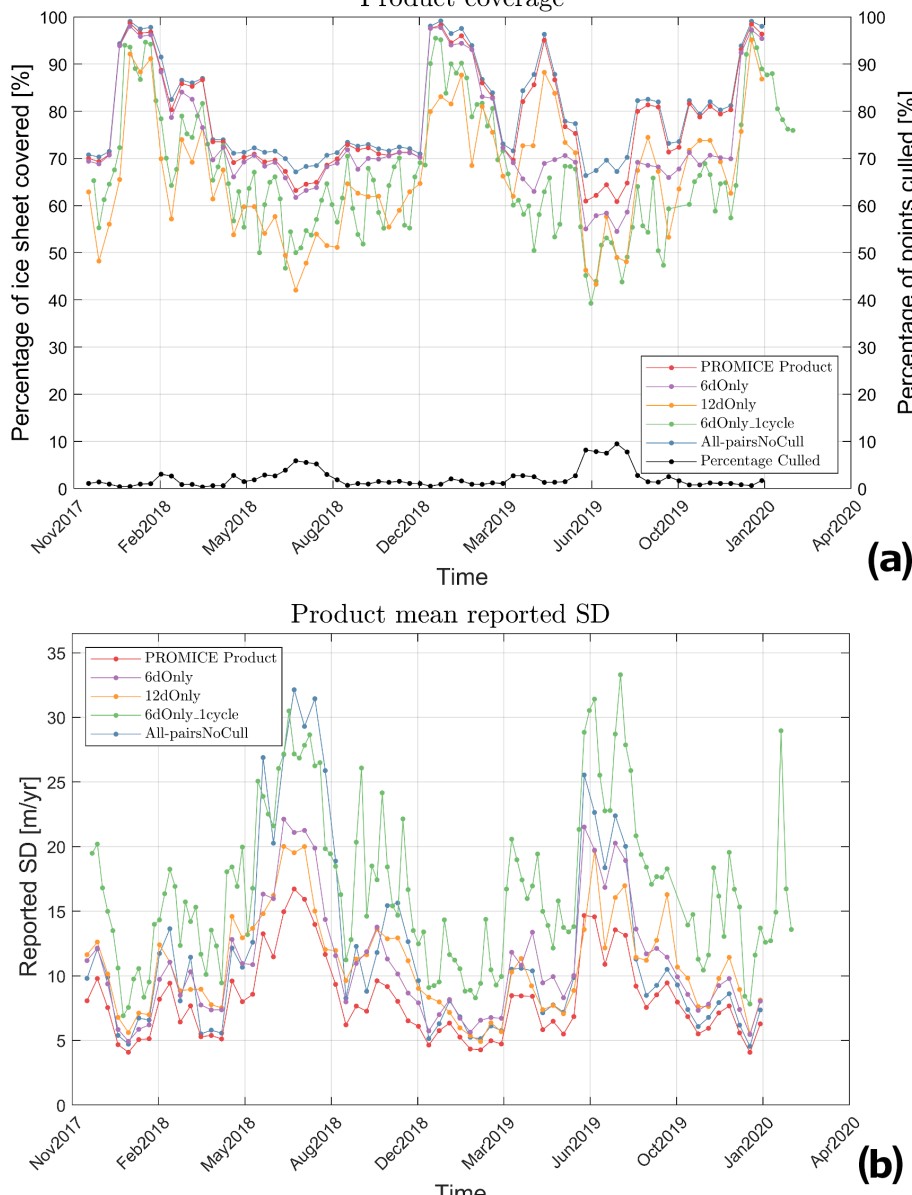

**Figure 10. (a)** The left axis shows the percentage of the ice sheet that is covered by data for each mosaic. Right axis: the lower blue curve shows the percentage of pixels that have been culled for the PROMICE product using the procedure described in Sec. 4.4. **(b)** The mean reported standard deviation for each mosaic.

Earth Syst. Sci. Data, 13, 1–22, 2021                https://doi.org/10.5194/essd-13-1-2021

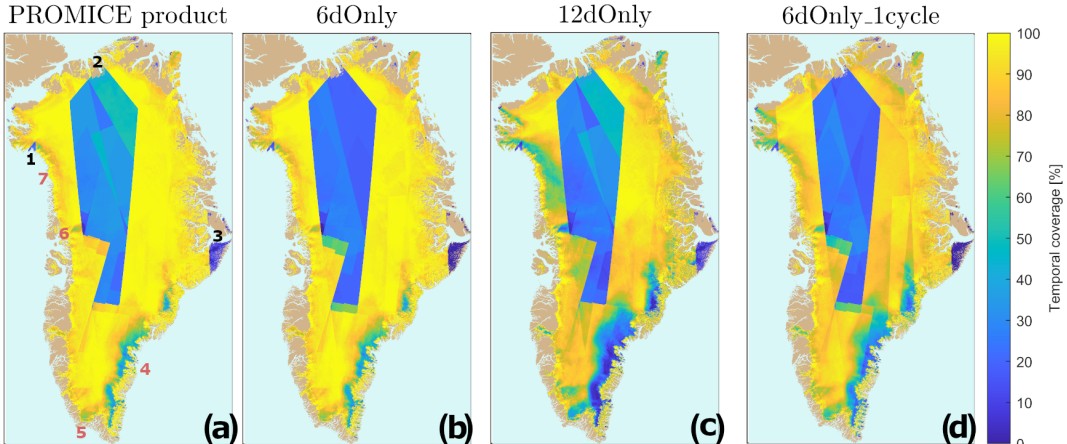

**Figure 11.** Temporal coverage: spatial view of the percentage of all mosaics that have data in a given grid point: **(a)** the PROMICE Ice Velocity product, **(b)** 6dOnly time series, **(c)** 12dOnly time series, and **(d)** 6dOnly_1cycle time series. The numbers in panel **(a)** indicate the locations of the areas mentioned in Sect. 6. Areas where SAR in IW mode has not been acquired on a regular basis: 1, 2, and 3 refer to the triangular area in Melville Bay, north Greenland, and the Scoresbysund area, respectively. Areas with low ice velocity coverage: 4, 5, 6, and 7 refer to the southeast ice sheet margin, small area in south Greenland, an area north of Rink Glacier, and the Melville Bay area, respectively.

In the PROMICE Ice Velocity product each mosaic includes all possible 6 and 12 d pairs within two consecutive Sentinel-1A cycles, and the timestamps supplied with the product lists the time span of the product (first and last date) as well as the midpoint time as specified in Table 1. This information is true for the mosaic but not for a given grid point. This is due to the following:

- The SAR data (Fig. 2) are not acquired simultaneously over the GrIS as described in Sect. 3.1.

- Different areas on the ice sheet are covered by a varying number of tracks/varying amount of data acquired at different times (also Fig. 2).

- Although data are acquired, the processor is unable to detect displacement for some pixels or larger areas due to loss of coherence, or the processing of an image pair fails for various reasons and is therefore not included in the final mosaic.

Another point to keep in mind is that the mosaic is a weighted average of the processed pairs. This means that, although the product has a high temporal resolution, the time series will be smoothed and likely miss short-lived (real) peaks in velocity, for instance during summer.

The analysis from this section shows not only that it is possible to provide a Greenland-wide ice velocity product with a higher temporal resolution than the PROMICE product (the 6dOnly_1cycle product) but also that this comes at the price of reduced spatial coverage and higher uncertainty. Creating a product spanning more than two Sentinel-1A cycles will have opposite effects. The choice of two Sentinel-1A cycles for the PROMICE product is therefore a compromise be-

tween having reasonably high temporal resolution and good coverage and reducing noise.

## 7 Validation

We validate the PROMICE Ice Velocity product against in situ GPS measurements from the PROMICE automatic weather stations (AWSs) (van As et al., 2011; Fausto and van As, 2019) and perform an analysis over stable ground. Only a limited number of GPS measurements are available since the data must overlap in time with the period of the PROMICE Ice Velocity product and have a temporal resolution comparable to or higher than the PROMICE Ice Velocity product. Furthermore, the measurements are biased toward the slow-moving parts of the ice sheet ablation zone. Locations are displayed in Fig. 12.

PROMICE AWSs measure a range of surface mass-balance components in the ablation zone of the GrIS. The stations are per design located in slow-moving areas with an average flow generally lower than $100 \, \text{m yr}^{-1}$ (Fig. 12). The position of the AWS is measured every hour using a single-frequency GPS receiver and a small ceramic patch active antenna. We use the freely available hourly positions (Fausto and van As, 2019) to calculate velocities using a workflow similar to that described in GIScci-Consortium (2018): daily positions of the GPS stations are calculated as a mean of the hourly positions for each day. The velocity components are estimated using a weighted linear regression for each of the 24 d time spans of the velocity mosaics using the daily positions. The weights are inversely proportional to the number of hourly measurements going into the estimate of a daily position in order to account for gaps in the data.

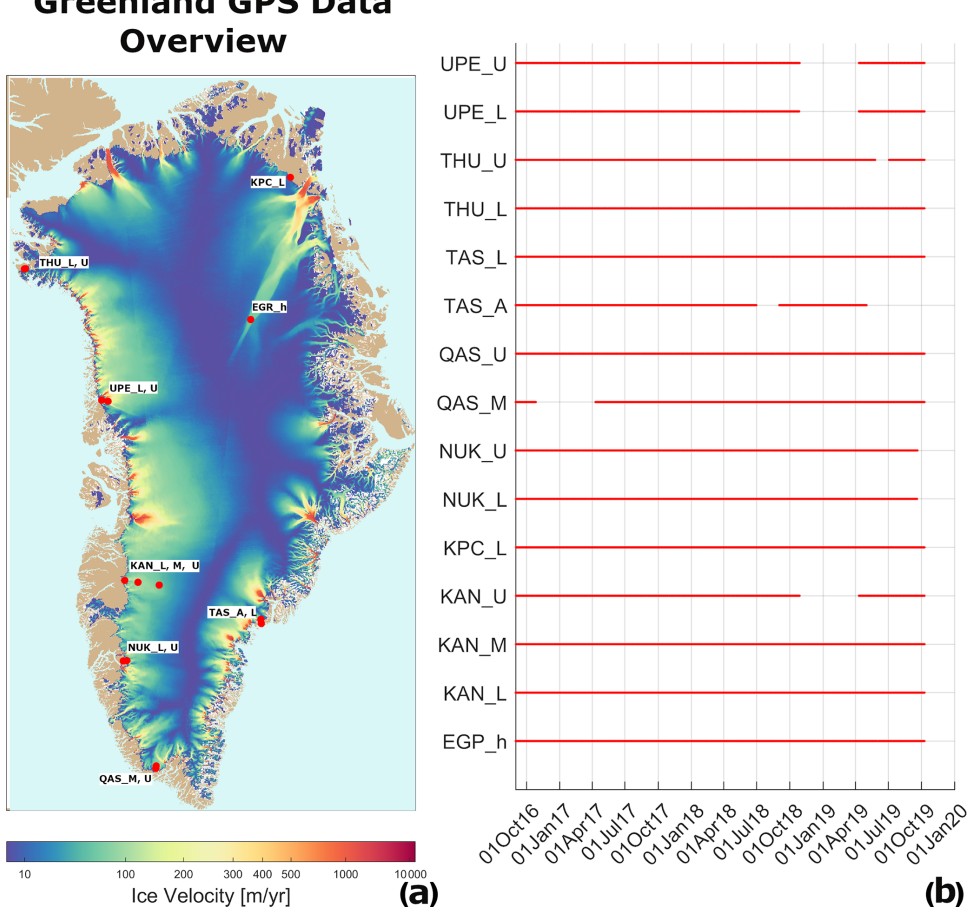

**Figure 12.** Overview of PROMICE GPS data: **(a)** locations of the PROMICE GPS stations on the GrIS. The ice velocity mosaic used as the base layer is a 3-year average of all the PROMICE Ice Velocity maps spanning September 2016 to September 2019. **(b)** List of PROMICE GPS stations used in the validation and their data coverage.

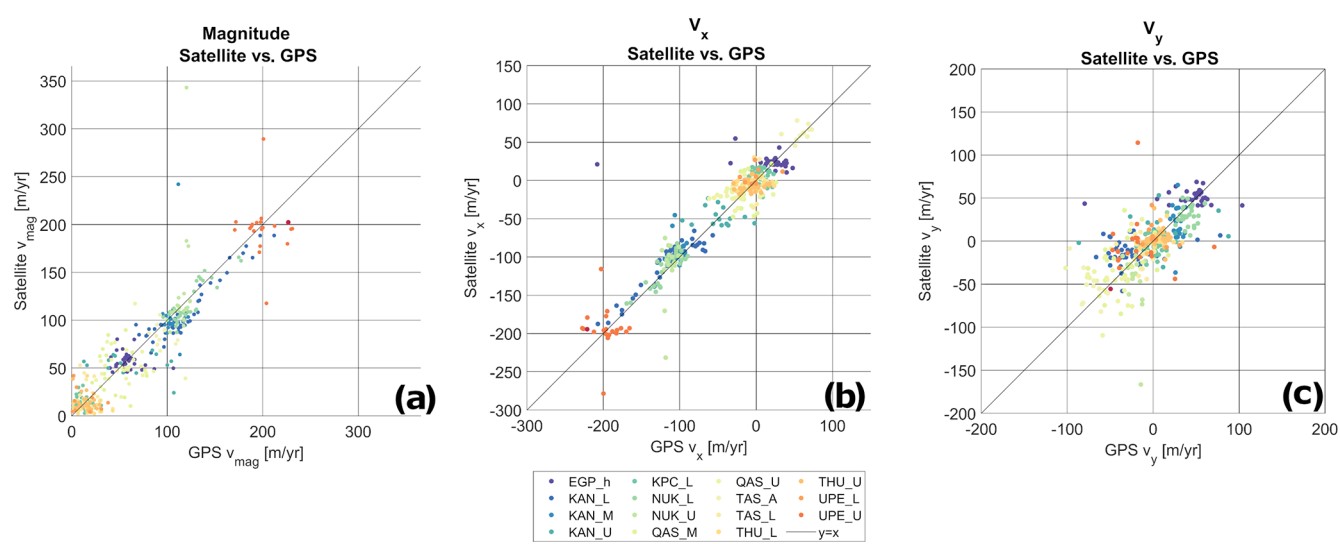

**Figure 13.** Scatter plots of PROMICE GPS ice velocity vs. PROMICE Ice Velocity. **(a)** Scatter plot of the magnitude of the velocity. **(b)** Scatter plot of the $v_x$ component. **(c)** Scatter plot of the $v_y$ component.

**Table 5.** Statistics of the validation of the satellite ice velocity products using PROMICE GPS data.

| Product | Magnitude | | $x$ dir | | $y$ dir | |
|---|---|---|---|---|---|---|
| | SD ($\mathrm{m\,yr^{-1}}$) | Bias ($\mathrm{m\,yr^{-1}}$) | SD ($\mathrm{m\,yr^{-1}}$) | Bias ($\mathrm{m\,yr^{-1}}$) | SD ($\mathrm{m\,yr^{-1}}$) | Bias ($\mathrm{m\,yr^{-1}}$) |
| PROMICE product (all pairs) | 19 | 4 | 20 | −3 | 27 | −2 |
| 6dOnly | 22 | 0 | 21 | −2 | 31 | −1 |
| 12dOnly | 19 | 4 | 20 | −2 | 28 | 0 |
| 6dOnly_1cycle | 34 | 9 | 31 | −3 | 40 | −2 |

Scatter plots of the satellite-derived PROMICE Ice Velocity product (magnitude, $v_x$ and $v_y$ components) vs. PROMICE GPS-derived ice velocities are displayed in Fig. 13. The standard deviation of the difference between the GPS measurements and the satellite-derived velocity (from here on referred to as the standard deviation) is calculated along with the mean difference (bias) between GPS and satellite velocity (see first line in Table 5). These values reflect not only the uncertainty of the satellite product but also that of the GPS-derived velocity. The expected error of the satellite product is estimated to be 10–30 $\mathrm{m\,yr^{-1}}$ for individual pairs (GIScci-Consortium, 2013) TS1 . The PROMICE product lies well within these bounds, with a standard deviation and bias of 20 and −3 $\mathrm{m\,yr^{-1}}$ for the $v_x$ component and 27 and −2 $\mathrm{m\,yr^{-1}}$ for the $v_y$ component, respectively. The larger standard deviation for the $v_y$ component is expected: due to the general north–south orientation of the satellite tracks (Fig. 2), the $v_y$ component is aligned roughly parallel to the azimuth direction (satellite flight path), and Sentinel-1 IW SLC images (see Sect. 3.1) have a much lower resolution in azimuth than in the range (line-of-)sight direction. We note that due to the roughly east–west orientation of most Greenland glaciers the velocity range of the $y$ component in our validation is notably smaller than that of the $x$ component.

We perform a similar analysis for the PROMICE product for the pixels over stable ground, where no movement is expected. All pixels on ice-free terrain from all mosaics (each spanning 24 d) in the time series are included, which totals more than $142 \times 10^6$ pixels. The resulting values of the standard deviation and bias are 8 and 0.1 $\mathrm{m\,yr^{-1}}$ for the $v_x$ component and 12 and −0.6 $\mathrm{m\,yr^{-1}}$ for the $v_y$ component, respectively (see Table 6). The values of the standard deviation are less than half the values of the validation against GPS measurements, while the biases are significantly lower (in absolute value) and thus closer to zero. In this analysis, the standard deviation and bias are also largest for the $v_y$ component as discussed above.

Hvidberg et al. (2020) carried out a validation of many available satellite-derived ice velocity products using an array of 63 GPS stations around East Greenland Ice-core Project (EastGRIP) deep-drilling site (75°38′ N, 35°60′ W) located on the Northeast Greenland Ice Stream. The East-

GRIP deep-drilling site is located in the ice-sheet accumulation zone, and Hvidberg et al. (2020) measured an average speed in the central flow line of 55 $\mathrm{m\,yr^{-1}}$. The ice thus flows slowly at this location. The PROMICE Ice Velocity product was included in this analysis, and they found an average standard deviation of 6.6 $\mathrm{m\,yr^{-1}}$ for the products within the period 13 September 2016 to 8 August 2019. This is a significantly lower value of the standard deviation compared to what the validation against the PROMICE GPS observations shows, but it is similar to the analysis carried out on stable ground in Sect. 5.2 (see Fig. 6). This difference is mainly due to two things: (1) in the accumulation zone, changes at the ice-sheet surface are most often due to snow fall and redistribution by wind. In contrast, in the ablation zone, where all the PROMICE GPS stations are located, the surface properties are influenced by several factors, e.g. melt, high accumulation rates, and rain. This influences the coherence of image pairs and thereby increases uncertainty in the velocity product in these areas. (2) The uncertainty of the GPS measurements reported in Hvidberg et al. (2020) is lower compared to the PROMICE GPS observations. This is due to both the longer temporal baseline between measurements and the data acquisition time of 2–4 h per data point. The velocities derived from the PROMICE GPS observations may therefore carry a non-negligible part of the uncertainty in the validation. None of the other products in the comparison by Hvidberg et al. (2020) have a similar high temporal resolution to the PROMICE Ice Velocity product. However, a 3-year average of the PROMICE Ice Velocity product was also included in the analysis, and Hvidberg et al. (2020) found that it had a standard deviation (0.7 $\mathrm{m\,yr^{-1}}$) similar to other offset/feature tracking products covering longer time spans like the annual maps from ESA CCI (Greenland Ice Sheet velocity maps from Sentinel-1 (Nagler et al., 2015)) and MEaSUREs Greenland Annual Ice Sheet Velocity Mosaics from SAR and Landsat, Version 1 (2015–2018) (Joughin et al., 2010), which have average values of 1.5 and 1.4 $\mathrm{m\,yr^{-1}}$, respectively (supplementary material in Hvidberg et al., 2020). The standard deviation of the 3-year average of the PROMICE product is lower than for the annual maps, most likely because it includes more data. This is also a conclusion drawn by Hvidberg et al. (2020).

**Table 6.** Statistics of the validation of the PROMICE product (spanning 24 d) over stable ground. Number of pixels included in the analysis: $> 142 \times 10^6$.

| Product | Magnitude | | $x$ dir | | $y$ dir | |
|---|---|---|---|---|---|---|
| | SD $(\mathrm{m\,yr^{-1}})$ | Bias $(\mathrm{m\,yr^{-1}})$ | SD $(\mathrm{m\,yr^{-1}})$ | Bias $(\mathrm{m\,yr^{-1}})$ | SD $(\mathrm{m\,yr^{-1}})$ | Bias $(\mathrm{m\,yr^{-1}})$ |
| PROMICE product (all pairs) | 10 | 10 | 8 | 0.1 | 12 | −0.6 |

The 12dOnly and 6dOnly time series introduced in Sect. 6 have similar standard deviations to the PROMICE product, when compared to the velocity derived from the PROMICE GPSs, whereas the higher-temporal-resolution product, 6dOnly_1cycle, has a significantly higher standard deviation. The PROMICE Ice Velocity product has the lowest standard deviation of the four. When using only 6 d pairs, it is also possible to define a Greenland-wide product with a temporal resolution of 12 d, – the 6dOnly_1cycle product. It has the clear advantage of resolving the dynamics of the outlet glaciers even better, although this comes at the price of increased noise due to both the shorter temporal baseline and the geolocation bias as well as reduced coverage of each mosaic (Figs. 10a and b and 11 and Table 5). For studies concerned with changes in fast flow the increased temporal resolution may outweigh the downsides.

The uncertainty reported in the ice velocity product is lower than the values we found during our validation against GPS. The average standard deviation found in Hvidberg et al. (2020) or the stable ground analysis is more comparable. The origin of some errors is such that the algorithm is unable to account for them. This is especially true for the spatially correlated errors caused by ionospheric scintillations (Sect. 5.4), which are not fully estimated by the error estimation algorithm (Sect. 5.6). A second issue is the distribution of the PROMICE AWSs biased towards the slow-flowing parts of the ablation zone as well as the uncertainty in the velocity estimates from these data.

For a time series of mosaics like the PROMICE Ice Velocity product, errors will vary both spatially and temporally due to the sources described in Sect. 5 as well as to variations in data coverage (Sect. 4.3 and Fig. 2a). A validation dataset which is not biased towards slow-flowing areas in the ablation zone but is representative of a larger range of flow regimes and surface conditions would help assess whether the reported product errors capture this correctly. The analysis above, however, shows that the size of the product errors are as expected.

## 8 Living data: updates and improvements

PROMICE will continue to distribute and update the PROMICE Ice Velocity product based on the Sentinel-1 data collected and released by ESA and the Copernicus programme. We aim to deliver a clean and homogenous data product and offer the possibility of user interaction and addressing issues with the data product. Associated with PROMICE, we have a user-contributable dynamic web-based data archive (GitHub), which lists known data quality issues (https://github.com/GEUS-PROMICE/Sentinel-1_Greenland_Ice_Velocity, last access: 1 July 2021). On the GitHub page, we also offer the opportunity for data users to add and document new issues. Documenting dataset issues is often simpler than correcting them, and future dataset versions will implement fixes to any verified issues as soon as they are done. All fixed issues will be tagged as closed and remain visible for new users.

We encourage users who are working with Sentinel-1 and the PROMICE Ice Velocity data to search the issue database and see if there are any known data issues relevant to their needs. We find it likely that there are issues unknown to us in the existing data, and new issues may be found in the future data collection pipeline. We will do our best to improve the dataset with user-based help through the GitHub page.

## 9 Data availability

The PROMICE Ice Velocity product has DOI https://doi.org/10.22008/promice/data/sentinel1icevelocity/ greenlandicesheet (Solgaard and Kusk, 2021) and is available at https://dataverse01.geus.dk/dataverse/Ice_velocity (last access: 1 July 2021). The product is updated regularly with a new mosaic every 12 d. Check out https://github.com/GEUS-PROMICE/Sentinel-1_Greenland_Ice_Velocity for updates and for posting issues. GPS measurements from PROMICE AWS stations are available at https://doi.org/10.22008/promice/data/aws TS2 (Fausto and van As, 2019; van As et al., 2011). Greenland Ice Sheet surface elevation data from the GIMP project are available from https://doi.org/10.5067/NV34YUIXLP9W TS3 (Howat et al., 2015, 2014). Grounding line data from the ESA CCI project are available at http://products.esa-icesheets-cci.org/products/downloadlist/GLL/ TS4 (last access: 1 July 2021).

## 10 Summary and outlook

We have presented the PROMICE Ice Velocity product – a time series of GrIS-wide velocity mosaics (September 2016 to present) based on Sentinel-1 SAR data. The product has

a 500 m spatial and 24 d temporal resolution and is produced in an operational setup using the IPP processor. A new mosaic is produced every 12 d and is made available within 10 d of the last included acquisition. During the winter campaigns, this lag is larger due to the amount of data to be processed. Validation against PROMICE AWS GPS data shows that the standard deviation of the difference between the ice velocity product and the GPS data is 20 and 27 m yr$^{-1}$ for the $v_x$ and $v_y$ component, respectively. This is within the expected uncertainty range of 10–30 m yr$^{-1}$ (GIScci-Consortium, 2013). However, we expect the actual values pertaining to the PROMICE Ice Velocity product to be lower as the PROMICE AWS GPS data carry a non-negligible part of the uncertainty. This is indicated by the analysis carried out for pixels over stable ground – which showed a standard deviation of 8 and 12 m yr$^{-1}$ for the $v_x$ and $v_y$ component, respectively – as well as by the study of Hvidberg et al. (2020). Better spatially distributed validation data with low uncertainty would help in assessing whether the processor captures the spatially and temporally varying uncertainty field correctly.

Ice velocities are retrieved by applying intensity offset tracking to Sentinel-1 images acquired 6 and 12 d apart. The resulting velocity maps from all image pairs acquired during a 24 d period are temporally averaged and mosaicked to produce a consistent coverage. The processing chain is described in detail from the input data to the final outlier removal. We discuss the various error sources, which include biases and smoothly varying errors due to orbit and timing errors, noise-like errors due to changes in radar backscatter between radar acquisitions, and errors due to ionospheric scintillations. The error estimation approach is also described.

We show how the product coverage varies temporally and spatially in response to variations in SAR data acquisitions and seasonal changes in surface properties. The southeast GrIS margin has good Sentinel-1 SAR data coverage but often has gaps in the mosaics due to changes in the surface properties caused by surface melt and high precipitation rates, which hinder velocity retrieval. Other areas, like the small triangular area in the Melville Bay area, have low coverage in the mosaics simply due to lack of SAR data.

The PROMICE Ice Velocity product will continue to update as long as the Sentinel-1 satellites are in operation. We will continue to make improvements to the product, and these updates will be posted at https://github.com/GEUS-PROMICE/Sentinel-1_Greenland_Ice_Velocity. Users are encouraged to add and document product issues or suggest improvements.

The PROMICE Ice Velocity product presented here was originally intended primarily to calculate ice discharge through marine-terminating glaciers of the GrIS as done in Mankoff et al. (2020). The PROMICE Ice Velocity product is thus less suited for studying very short lived changes in the velocity structure, as observed in situ by e.g. Bartholomew et al. (2012) and Ahlstrøm et al. (2013) or through higher-frequency acquisitions/non-mosaic products of satellite imagery as done by e.g. Sundal et al. (2013) and Davison et al. (2020). By not mosaicking all the individual image pairs like we do for the PROMICE Ice Velocity product, a much higher temporal resolution over a limited region is possible. Yet the spatially comprehensive and temporally consistent nature of the PROMICE Ice Velocity product also makes it attractive for longer-term large-scale monitoring of the GrIS velocity structure and glacier dynamics as done by Vijay et al. (2019) and Solgaard et al. (2020).

**Author contributions.** AS and AK designed and produced the PROMICE Ice Velocity product. AK, JPMB, and JD developed the processing software. RSF and KDM set up the data-curation framework. AS and AK prepared the manuscript with contributions from APA, JPMB, SBA, NBK, SHL, MC, NJK, and KKK.

**Competing interests.** The authors declare that they have no conflict of interest.

**Disclaimer.** Publisher's note: Copernicus Publications remains neutral with regard to jurisdictional claims in published maps and institutional affiliations.

**Special issue statement.** This article is part of the special issue "Extreme environment datasets for the three poles". It is not associated with a conference.

**Acknowledgements.** The authors thank the editor and Ben Davison as well as two anonymous reviewers for constructive comments and feedback. Ice velocity maps were produced as part of the Programme for Monitoring of the Greenland Ice Sheet (PROMICE) using Copernicus Sentinel-1 SAR images distributed by ESA and were provided by the Geological Survey of Denmark and Greenland (GEUS) at http://www.promice.org (last access: 1 July 2021). AWS data from PROMICE and the Greenland Analogue Project (GAP) were provided by GEUS at http://www.promice.org. Grounding line data were provided by the ESA Greenland Ice Sheet CCI project (https://climate.esa.int/en/projects/ice-sheets-greenland/, last access: 1 July 2021).

**Financial support.** PROMICE is financed by the Ministry of Climate, Energy and Utilities through the climate support programme DANCEA (Danish Cooperation for Environment in the Arctic), which is managed by the Danish Energy Agency.

**Review statement.** This paper was edited by Xin Li and reviewed by Ben Davison and two anonymous referees.

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

## Remarks from the typesetter