# Peer review of "Greenland ice velocity maps from the PROMICE project"

_Earth System Science Data, 2021_

## Referee Comment (RC1)

**Review of "Greenland ice velocity maps from the PROMICE project" by Anne Solgaard et al.**

This manuscript entitled "Greenland ice velocity maps from the PROMICE project" generates a time series of Greenland Ice Sheet (GIS) velocity mosaic spanning from September 2016 to present with a high temporal resolution of 24 days and a spatial resolution of 500 m based on the Sentinel-1 SAR data. In the main text, the authors introduced the data processing steps (e.g., how to extract the ice velocity and how to perform the data fusion) and the relevant error analysis (e.g., regarding the orbit error, geolocation bias correction, ionospheric effects etc.) and the validation of the results (by comparison with the in-situ GPS measurements) in detail. Overall, this manuscript was well written and organized, and the produced dataset is reliable and acceptable. And the reviewer believes that the dataset will largely contribute to the understanding of GIS ice dynamics and the behind mechanism of accelerated mass loss. Hence, this manuscript can be accepted once some issues listed below are addressed.

**General comments:**

1. With regard to the extraction of the glacier velocity when using the offset-tracking technique, the issue of how to perform the co-registration for huge amounts of SAR data, which is the most important steps for remote sensing data processing, is left out. Please clarify relevant issues, and a more detailed description is preferred.

2. For the error assessment of the final glacier velocity dataset, the offsets in ice-free areas are generally evaluated and discussed. However, this manuscript did not give relevant explanations and discussions. This information needs to be further described.

3. Some issues with respect to the formatting and written need to be revised, please see the specific comments.

**Specific comments:**

P1: In the abstract, after the location of the website, the authors added the references (Solgaard and Kusk, 2021), which corresponds to the content of this manuscript. Please confirm if this is OK?

P1, L9: Suggest that using the phrases "north-south direction" and "east-west direction" replaces the Vx and Vy.

P4, L4-6: The "Fig. 2A" and "Fig. 2B" should be changed to "Fig. 2a" and "Fig. 2b", respectively.

P4, L4-6: Changing the phrase "See 5" to "See section 5" seems to be better. There are

several errors of this type in the main text, please check and fix them.

P5, L7-8: More information about the GIMP DEM is needed.

P5, L10: Does the term ("TPP") have a full name? if so, please clarify.

P6: For the flow chart (i.e., Fig.3), it is just a simple and conventional description for the data processing of this study. The reviewer suggests that more detailed processes of data processing should be given in this figure. In addition, add the item related to the data co-registration.

P6, L7-9: Please add the word "section" to these expressions like "in 4.2", "in 4.3 and 5", "in 4.4".

P7: For the section 4.2 ("Offset-tracking"), please add and clarify the issue of how to perform data co-registration (more information needs to be given), which is one of the most important procedures when employing the offset-tracking technique to extract the glacier velocity. In particular, for the constantly updated data, how to deal with?

P10, L15: This expression like "the range standard deviation" can easily be misunderstood. So a more clear expression like "the standard deviation in range direction" would be better. Similar issues can also be seen in Table 2, e.g., "Rang is line of sight and azimuth is along the satellite flight path".

P13, L8: Please confirm if the units are correct? (9.7 m/s and 24.4 m/s?)

P15, L8-10: From the magnitude of the error (up to 300 m/a) caused by ionospheric effects, this item is the most significant uncertainty source. Why didn't the authors try to apply the relevant methods mentioned in last paragraph? The reviewer thinks that developing a better or targeted method is necessary, as the author also mentioned that the post-processing step did not completely eliminate this type of error.

P15: In the section 5.5, when it comes to the error assessments, a common method of calculating the offsets over the ice-free areas is usually adopted and further analyzed. So relevant information regarding the results of ice-free areas deserved to be given.

P15, L16-17: Providing a clear explanation regarding the correct magnitude of the errors seems to be better, now that you mentioned the study of Boncori et al. (2018). Moreover, the format of the reference "in (Boncori et al., 2018)" can be changed to "in Boncori et al. (2018)". Please check.

P16: To ensure the consistency in writing, the word "Primice" in the subtitle should be changed to "PRIMICE".

P17, L4-6: Please add the position information about the Melville Bay and Scoresbysund areas in Fig. 2.

P23: For the Fig. 12 (also for Fig. 4), a revised figure with higher resolution is needed. Now it looks a little fuzzy.

---

## Author Comment (AC1)

Dear Editor,

We hereby submit our reply to the three reviews on our manuscript, "Greenland ice velocity maps from the PROMICE project" by Solgaard et al.

In the following, we provide a point-by-point response to all the reviewer comments. The reviewer comments are in black while our replies are in green using a different font for clarity. A tracked changes version of the manuscript is included.

We have followed all suggestions in the reviewers' general comments except one. Reviewer #3 suggests that we compare our velocity products against other already available satellite-derived velocities. While this is a worthwhile exercise which would be of interest to the large community of users, we find that it is out of scope for our manuscript. We have performed a thorough validation of our product using available in-situ GPS observations and as suggested by both Reviewer #1 and #2 included an analysis of pixels covering stable ground. Both analyses as well as the study by Hvidberg et al, 2020 show that the PROMICE product performs as expected.

We thank the editor and all three reviewers for useful and constructive feedback and comments.

Sincerely,

Anne Solgaard on behalf of all co-authors.

**REVIEWER 1**

Review of "Greenland ice velocity maps from the PROMICE project" by Anne Solgaard et al. This manuscript entitled "Greenland ice velocity maps from the PROMICE project" generates a time series of Greenland Ice Sheet (GIS) velocity mosaic spanning from September 2016 to present with a high temporal resolution of 24 days and a spatial resolution of 500 m based on the Sentinel-1 SAR data. In the main text, the authors introduced the data processing steps (e.g., how to extract the ice velocity and how to perform the data fusion) and the relevant error analysis (e.g., regarding the orbit error, geolocation bias correction, ionospheric effects etc.) and the validation of the results (by comparison with the in-situ GPS measurements) in detail. Overall, this manuscript was well written and organized, and the produced dataset is reliable and acceptable. And the reviewer believes that the dataset will largely contribute to the understanding of GIS ice dynamics and the behind mechanism of accelerated mass loss. Hence, this manuscript can be accepted once some issues listed below are addressed.

**General comments:**

1.With regard to the extraction of the glacier velocity when using the offset-tracking technique, the issue of how to perform the co-registration for huge amounts of SAR data, which is the most important steps for remote sensing data processing, is left out. Please clarify relevant issues, and a more detailed description is preferred.

No resampling is carried out to coregister the images, for several reasons:
(a) The ice motion is spatially variant, and can represent several pixels of displacement. To coregister accurately, it would thus have to be known a priori, but since the displacement is the quantity we wish to measure, this is not feasible.
(b) Intensity cross-correlation does not rely on the complex image phase, and employs large windows (256x64 in this case), therefore subpixel resampling of the images is not necessary. Instead we just extract patches in the images centered on integer pixel locations, computed based on precise state vectors and on the acquisition geometry, and correct subsequently for the fractional pixel shift.

We have revised section 4.2, and added the following clarification regarding coregistration:
*"For intensity cross-correlation methods, the SLCs need not be coregistered and resampled to sub-pixel accuracy prior to the processing, as they rely on relatively large windows (several tens of pixels in each dimension). Instead, for the regular grid points in the reference SLC, we calculate the expected position of the corresponding grid points (assuming no motion) in the second SLC, based solely on SLC timing information, orbital state vectors, and the DEM. The grid points are selected on integer pixel positions in the reference SLC, but the corresponding grid points will generally not coincide with integer pixel locations in the second SLC. To avoid resampling the second SLC, we round the corresponding grid points*

*to their nearest integer pixel locations and save the fractional shifts, which are then added back to the offset measurement after cross-correlation."*

2.For the error assessment of the final glacier velocity dataset, the offsets in ice-free areas are generally evaluated and discussed. However, this manuscript did not give relevant explanations and discussions. This information needs to be further described.

Yes, we completely agree. This important measure of product performance. We have carried out an analysis of all ice free pixels in all the mosaics of the PROMICE product (more than $142 \cdot 10^6$ points). This analysis gives:

|  | vmag | vx | Vy |
|---|---|---|---|
| STD | 9.8 m/yr | 7.7 m/yr | 11.9 m/yr |
| Mean of all points | 10.2 m/yr | 0.1 m/yr | -0.6 m/yr |
| # of points | 142710462 | 142710462 | 142710462 |

We have inserted a paragraph following the validation against GPS describing the analysis of stable ground pixels in Section 7: Validation:

*'We perform a similar analysis for the PROMICE product for the pixels over stable ground, where no movement is expected.All pixels on ice-free terrain from all mosaics (each spanning 24 days) in the time series are included, which totals to more than $142 \cdot 10^6$ points. The resulting values of the standard deviation and bias 8 m/yr and 0.09 m/yr for the $v_x$-component and 12 m/yr and -0.6 m/yr for the $v_y$-component, respectively (see Tab. 6). The values of the standard deviation are less than half the values of the validation against GPS measurements, while the biases are significantly lower (in absolute value) and thus closer to zero. In this analysis, the standard deviation and bias are also largest for the $v_y$-component as discussed above.'*

A table has also been included for comparison to the validation using GPS-observations: Table 6:

| Product | Magnitude | | x-dir | | y-dir | |
|---|---|---|---|---|---|---|
|  | Std | Bias | Std | Bias | Std | Bias |
|  | [m/yr] | [m/yr] | [m/yr] | [m/yr] | [m/yr] | [m/yr] |
| PROMICE Product (All pairs) | 10 | 10 | 8 | 0.1 | 12 | -0.6 |

3.Some issues with respect to the formatting and written need to be revised, please see the specific comments.

**Specific comments:**

P1: In the abstract, after the location of the website, the authors added the references (Solgaard and Kusk, 2021), which corresponds to the content of this manuscript. Please confirm if this is OK?

We confirm this. The DOI and dataset citation is included in accordance with the manuscript guidelines:
1. Abstract: the abstract should be intelligible to the general reader without reference to the text. After a brief introduction of the topic, the summary recapitulates the key points of the article and mentions possible directions for prospective research. Reference citations should not be included in this section (except for data sets) and abbreviations should not be included without explanations. At least for the final accepted publication, a functional data set DOI and its in-text citation must be given in the abstract. If multiple data set DOIs are necessary, please instead refer to the data availability section.

P1, L9: Suggest that using the phrases "north-south direction" and "east-west direction" replaces the Vx and Vy.
Agree, this makes good sense in the abstract. It now reads:
'...for the components in the eastern and northern direction, respectively.'

P4, L4-6: The "Fig. 2A" and "Fig. 2B" should be changed to "Fig. 2a" and "Fig. 2b", respectively.
Done

P4, L4-6: Changing the phrase "See 5" to "See section 5" seems to be better. There are several errors of this type in the main text, please check and fix them.

They are fixed.

P5, L7-8: More information about the GIMP DEM is needed.

We have added:

'We employ the Greenland Ice Mapping Project (GIMP) DEM based on the ASTER and SPOT 5 DEMs and AVHRR photoclinometry (Howat et al., 2014, 2015), downsampled to 500 m spacing to match the resolution of the IV product.'

And included the missing dataset reference: Howat et al., 2015.

P5, L10: Does the term ("TPP") have a full name? if so, please clarify.

We have added:

"The data processing is carried out using the Interferometric Post Processing (IPP) processor, developed and maintained by DTU Space \citep{Kusk2018}. Despite the name, the processor also performs offset-tracking for displacement measurements, which is the functionality used to generate the PROMICE product"

P6: For the flow chart (i.e., Fig.3), it is just a simple and conventional description for the data processing of this study. The reviewer suggests that more detailed processes of data processing should be given in this figure. In addition, add the item related to the data co-registration.

The figure has been split in two sub-figures, with figure b showing a detailed flow diagram of the offset-tracking.

P6, L7-9: Please add the word "section" to these expressions like "in 4.2", "in 4.3 and 5", "in 4.4".

This has been fixed.

P7: For the section 4.2 ("Offset-tracking"), please add and clarify the issue of how to perform data co-registration (more information needs to be given), which is one of the most important procedures when employing the offset-tracking technique to extract the glacier velocity. In particular, for the constantly updated data, how to deal with?

This is addressed above in the answer to the first comment.

P10, L15: This expression like "the range standard deviation" can easily be misunderstood. So a more clear expression like "the standard deviation in range direction" would be better. Similar issues can also be seen in Table 2, e.g., "Rang is line of sight and azimuth is along the satellite flight path".

Agreed. We have changed the paragraph as the following

"We note that the range velocity bias (-0.5~m/yr) and azimuth velocity bias (0.5~m/yr) do not differ between the precise and the restituted orbit files. The standard deviation in the range direction is 2.7~m/yr for the precise orbit files and 2.8~m/yr for the restituted orbit files, while the standard deviation in the azimuth direction is 3.3~m/yr for both orbit types."

The caption for Table 2 has been corrected to "Range direction is line of sight and azimuth direction is along the satellite flight path". The headings in the Table have, for clarification, also been changed to "Range Velocity Bias / Range Velocity Std/ Azimuth Velocity Bias/Azimuth Velocity Std"

P13, L8: Please confirm if the units are correct? (9.7 m/s and 24.4 m/s?)

This was indeed a typo. The units have been changed to m/yr.

P15, L8-10: From the magnitude of the error (up to 300 m/a) caused by ionospheric effects, this item is the most significant uncertainty source. Why didn't the authors try to apply the relevant

methods mentioned in last paragraph? The reviewer thinks that developing a better or targeted method is necessary, as the author also mentioned that the post-processing step did not completely eliminate this type of error.

The ionospheric effects are indeed the major error source, and we have experimented with methods relying on the dispersive properties of the ionosphere, similar to the ones described in (Gomba 2018) and (Liao et. al., 2018). However for Sentinel-1 level 1 products, there is a problem with block processing artifacts present in most Sentinel-1 SLCs, which severely hampers this approach. Another method to reduce the impact of ionospheric effects is to exploit the fact that in some regions, measurements from both ascending and descending tracks are available, and in this case one can derive the horizontal velocity from only the range offsets, which are much less affected by the ionosphere. This is being worked on, and will be included in a future update of the processor. We have added the following text:

*"Another method to reduce the impact of ionospheric effects is to exploit the fact that in some regions, measurements from both ascending and descending tracks are available, and in this case, ice velocities can be derived from only the range offsets -- which are much less sensitive to ionospheric effects -- and the SPF assumption(see Sect.4.3). In the standard S1 acquisition plan (Fig. 2a), only two ascending tracks are acquired (the long track along the west coast of Greenland, and the track covering the northeast margin of the ice sheet up to the northernmost point), so the method will not be applicable everywhere. During the winter campaigns ( (Fig. 2b), this method will be applicable in a much larger part of the ice sheet. Work is undergoing to include this method in a future update of the PROMICE product."*

P15: In the section 5.5, when it comes to the error assessments, a common method of calculating the offsets over the ice-free areas is usually adopted and further analyzed. So relevant information regarding the results of ice-free areas deserved to be given.

We agree on this. Please see our response to comment nr.2 under General Comments. We have inserted a stable ground analysis in Section 7: Validation following the paragraphs on validation against GPS observations.

P15, L16-17: Providing a clear explanation regarding the correct magnitude of the errors seems to be better, now that you mentioned the study of Boncori et al. (2018). Moreover, the format of the reference "in (Boncori et al., 2018)" can be changed to "in Boncori et al. (2018)". Please check.

The reference format has been corrected.

P16: To ensure the consistency in writing, the word "Primice" in the subtitle should be changed to "PRIMICE".

Agree, done.

P17, L4-6: Please add the position information about the Melville Bay and Scoresbysund areas in Fig. 2.

We have added the locations of the areas discussed in Section 6 on Figure 10 and added a reference to the figure at the position in text. We used Figure 10 rather than Figure 2, because it is closer to the discussion in the manuscript.

P23: For the Fig. 12 (also for Fig. 4), a revised figure with higher resolution is needed. Now it looks a little fuzzy.

We have added less transparent grid lines to Figure 12 and increased the font size on the axes making it less fuzzy. The resolution of Figure 4 has been increased and the position of the grounding line has been added.

**REVIEWER 2: Ben Davison**

**General comments**

The PROMICE ice velocity product represents a significant improvement on existing operational Greenland Ice Sheet velocity products, particularly in terms of temporal resolution and timeliness. Indeed, the community has no doubt already benefitted substantially from this product and will continue to do so in future. With that in mind, this paper is an important source of documentation for what is, and will continue to be, a widely used product; therefore, it is crucial that the paper is clear and thorough throughout.

This paper presents a detailed description of the PROMICE ice velocity product and the operational processing chain used to produce it. The paper is (in most places) clear, detailed and thorough: it includes clear and accurate descriptions of each processing stage, error sources and estimation, validation, as well as the product itself. Whilst there are no major issues with the paper, I felt that some sections would benefit from more detail (or alternative descriptions) and, particularly in the second half of the manuscript, the writing lost some of the precision and concision found in the first half of the manuscript. I also have some suggestions that I believe will improve the clarity of the figures and associated descriptions. With these changes and suggestions implemented, I would be happy to recommend the manuscript for publication in this journal.

Below, I provide more specific comments, going through the paper line by line:

Page 1, line 7: when describing the GPS validation in the abstract, I think it would be beneficial to also provide the biases, and perhaps also the uncertainty estimation from velocity estimates over bedrock, because both are useful measures and ones which many readers will be familiar with.

Agree. We have carried out the analyses over stable ground and include the results in the abstract along with values for the bias (also for the validation with GPS):

'The product is validated against in-situ GPS measurements. We find that the standard deviation of the difference between satellite and GPS derived velocities (and bias) is 20 m/yr (-3 m/yr) and 27 m/yr (-2 m/yr) for the components in the eastern and northern direction, respectively. Over stable ground the values are: 8 m/yr (0.1 m/yr) and 12 m/yr (-0.6 m/yr) in the eastern and northern direction, respectively.'

We also include the analysis over stable ground in the validation section. See our response to Reviewer #1's General Comment #2.

Page 1, line 9: would 'east-west and north-south' be clearer than 'vx and vy' at this stage

in the manuscript?

Yes, this makes more sense in the abstract. See wording in the response to your previous comment.

Page 1, line 11: By 'excellent data coverage', do you mean comprehensive/complete imaging of the ice sheet by Sentinel-1? I realise that's a bit more clunky, but could you be more specific here?

Yes, it is more precise. The text now reads:
'Best spatial coverage is achieved in winter due to the comprehensive data coverage by Sentinel-1 and high coherence, while summer mosaics have the lowest coverage due to widespread melt.'

Page 1, line 14: 'and dynamics of glaciers' - can you add a timescale here? Such as '…over seasonal and longer timescales' (as discussed in the conclusion).

Yes, the last line now reads:
''The spatial comprehensiveness and temporal consistency make the product ideal for both monitoring and for studying ice-sheet wide ice discharge and dynamics of glaciers on seasonal scales.'

Page 1, line 18: I'd suggest that 'obtain ice sheet-wide observations of ice-flow velocities' makes an even stronger case for the product.

Agree, 'ice-sheet wide' is now included.

Page 1, line 22: Can you add 'tidewater glacier' before 'ice-flow velocities'? I assume this is the intended focus given the introduction on sea level rise and the reference (Ahlstrom et al., 2013). Alternatively, it would be worth also referencing one or two of the GPS studies carried out on land-terminating sectors.

We wanted to highlight that there just are not very many in-situ measurements of ice flow and especially not long term ones. We thus agree that the old wording was not clear on this, we have included references to more in-situ observations and changed the wording:

'In-situ measurements of ice-flow velocities are relatively sparse on the GrIS and most of the measurements stem from GPS surveys (Ahlstrøm et al., 2013).'
->
'In-situ measurements of ice-flow velocities (e.g. from GPS) are relatively sparse on the GrIS and are often of short duration (months) at high temporal resolution (e.g. Sole et al., 2011; Maier et al., 2019) or of longer duration (years) but at low temporal resolution (e.g. Thomas et al., 1998; Hvidberg et al., 2020) while few span several years at high temporal resolution (e.g. Ahlstrøm et al., 2013). '

Page 1, line 23: I suggest 'inaccessibility and size of the GrIS, as well as the harsh…'. Alternatively, a subsequent sentence stating that the size of the GrIS makes it impractical

to obtain continuous measurements of the whole ice sheet through field observations.

Yes, this is an important point. We go with your first suggestion.

Page 2, line 3: 'IV' doesn't seem to be used in preference to 'ice velocity' throughout the manuscript. I don't have a strong preference for either, but perhaps just stick to ice velocity for simplicity?

Yes, the abbreviation is unnecessary in the paper. We stick to 'ice velocity' throughout.

Page 2, line 28: The statement about weighted averaging is repeated a few lines later – perhaps it could be removed from one of the two sentences?

Agreed. The first occurrence has been changed to "each mosaic is based on velocity measurements from all possible 6- and 12-day pairs".

Page 4, line 2: I wonder if it would be worth providing the pixel spacing (2.3x14.1 m) as well as the resolution here? This would help maintain consistency with the description of the output grid dimensions in section 4.2, which seem to be based on pixel spacing.

This is correct, we have added "The pixel spacing of the product is 2.3 m in slant range, and 14.1 m in azimuth."

Page 4, line 20: Isn't stripmap mode higher resolution?

Yes, this is correct, the reference to product type here refers to the processing type (SLC/GRD/…), with IW acquisition mode implied. We have rephrased to "the highest resolution of the available IW product types".

/( check which resolution of DEM: Use GIMPdem90m) Page 5, line 8: Is the DEM downsampled before calculating the vertical component of ice
displacement? If so, I think it would be worth quantifying how this affects your vertical displacement estimate.

The DEM, when used for deriving the surface gradient in the surface parallel flow approximation, should approximately match the resolution of the offset maps, in order to avoid aliasing of high frequency slope variation. We have added the following to section 4.3:
*"The effective resolution of the velocity maps is on the order of the correlation window size, which corresponds to approximately 800×900 m on ground (see Sect. 4.2), so the resolution of the surface gradient map should approximately match this. The DEM is downsampled to the pixel spacing of the PROMICE product (500×500 m), and the gradient is derived using second order differences, which means the gradients are derived using samples approximately 1000 m apart."*

Page 6: The need to focus SAR images is mentioned briefly in the discussion, but I can't see any description of how the SAR images were focused, which should acknowledge the difficulties associated with TOPS mode data.

Focusing TOPS images is indeed tricky, which is why we use the SLC product, which is delivered focused by ESA using the Sentinel-1 Instrument Processing Facility (IPF). We have emphasised in Section 2.3 that the SLC images are already delivered as focused SAR images.

Page 7, lines 10-15: Was there any filtering of the intensity images prior to crosscorrelation to e.g. minimise the visibility of long-wavelength features, or to enhance the contrast in the images? I've found it increases the signal to noise ratio in my own, much smaller scale, investigations.

We do not apply a high-pass filter. The use of such filters are described in (deLange,2007), where it is applied successfully to intensity tracking on outlet glaciers. It is definitely of interest to us, but It is not clear, however, that it would be beneficial in the interior, where the offset-tracking relies on the presence of speckle, rather than features, or that the same filter parameters would be optimal everywhere. For the operational PROMICE product, where a manual fine-tuning is not feasible, we currently choose not to rely too much on adaptive, data-dependent processing, in order to have a consistent product.

Page 7, lines 10-15: I think it would be worth mentioning here that several techniques that are often used to co-register Sentinel 1 images fail over moving ice (such as crosscorrelation or Enhanced Spectral Diversity) fail because they rely on stationary surfaces, and so you have to rely on the orbit information and DEM.

Yes, this is indeed correct. We have addressed this also in an answer to reviewer 1, which I repeat below:
No resampling is carried out to coregister the images, for several reasons:

(c) The ice motion is spatially variant, and can represent several pixels of displacement. To coregister accurately, it would thus have to be known a priori, but since the displacement is the quantity we wish to measure, this is not feasible.

(d) Intensity cross-correlation does not rely on the complex image phase, and employs large windows (256x64 in this case), therefore subpixel resampling of the images is not necessary. Instead we just extract patches in the images centered on integer pixel locations and correct subsequently for the fractional pixel shift.

We have revised section 4.2, and added the following clarification regarding coregistration:
*"For intensity cross-correlation methods, the SLCs need not be coregistered and resampled to sub-pixel accuracy prior to the processing, as it relies on relatively large windows (several tens of pixels in each dimension). Instead, for the regular grid points in the reference SLC, we calculate the expected position of the corresponding grid points (assuming no motion) in the second SLC, based solely on SLC timing information, orbital state vectors, and the DEM. The grid points are selected on integer pixel positions in the reference SLC, but the corresponding grid points will generally not coincide with integer pixel*

*locations in the second SLC. To avoid resampling the second SLC, we round the corresponding grid points to their nearest integer pixel locations and save the fractional shifts, to be added back to the shift measurement after cross-correlation."*

Page 7, line 23: 'local medians' – can you specify what data this relates to? (i.e. velocity, flow direction etc)

The approach is applied independently to the range and azimuth velocities. We have included a expanded description of the approach:
*"Then, a further culling, is carried out on the on the range and azimuth shifts, using a normalized median test, as described in (Westerweel and Scarano, 2005).For each measurement,$U_0$, in a 5×5 neighbourhood, the median,$U_m$, of the 24 surrounding measurements($U_1,U_2,...,U_{25}$) is calculated (excluding $U_0$), and for each measurement in the neighbourhood, a residual,$R_i=|U_i-U_m|$is calculated. The median,$R_m$, of($R_1,R_2,...,R_{24}$) is then calculated, and used to normalize the residual of$U_0$ so that $R_m'=|U_0-U_m|/(R_m+\epsilon)$,where $\epsilon$ is a minimum normalization level that accounts for cross-correlation noise. We use $\epsilon= 0.1$ pixel, as suggested in(Westerweel and Scarano, 2005), and cull the measurement,$U_0$, if $R'_0$ exceeds a threshold of 5 for either of the range or azimuth shifts. This value was found by experiments to remove most clearly visible outliers, without removing valid measurements.Lower values removed more outliers, but had an adverse effect on measurement coverage"*

Page 7, line 24: I wonder if these outliers due to surface melt could be removed using a 'dusting' approach as in Selley et al (2021), or using a region growing approach as in Luttig et al. (2017)

Luttig, C., Neckel, N., and Humbert, A. 2017. A combined approach for filtering ice surface velocity fields derived from remote sensing methods. Remote Sensing, 9(10). DOI: https://doi.org/10.3390/rs9101062

Selley, H. L., et al. 2021. Widespread increase in dynamic imbalance in the Getz region of Antarctica from 1994 to 2018. Nature Communications, 1133. DOI: https://doi.org/10.1038/s41467-021-21321-1.

We believe the approach described in the answer to the previous question, in combination with the temporal culling of the final velocity maps, is very similar to the "dusting" approach described in (Selley, et.al., 2021). Like them, we also remove small unconnected segments of pixels in each offset map prior to fusion, as these tend to be unreliable. We have added the following sentence to Sect.4.2, after the expanded description of the median-based culling approach:
*"After the culling, small unconnected segments of pixels (<25 pixels) are removed, as these were found to often contain erroneous values."*

Page 7, last line: Is this technically a resampling? I thought it was actually a scattered interpolation, because the ground-surface pixel size varies throughout the image.

This is indeed a scattered interpolation. The definition of resampling seems to differ between various fields; in SAR literature, the term resampling is often used also for the irregular interpolation in slant-to-ground range conversion and geocoding. For clarity, we have changed the term "resampled" to "interpolated"

Page 8, section 4.4: When fusing velocity estimates from both 6- and 12-day pairs to generate the 24-day mosaics, are the different time periods considered? And how? I imagine you could interpolate them both to daily values and weight them accordingly.

This is described in section 4.3: "The shifts and standard deviations are converted to velocity by multiplying with the SLC pixel spacing and dividing by the temporal baseline". As the standard deviations (in the form of the inverse of the variances) are used to weight the measurements in the fusion (Eq. 3), this accounts for the varying temporal baselines.

Page 10, line 2: 'knowledge' seems an odd word to use here. Perhaps 'accuracy' would be better?
We have changed the sentence to. "Errors in the Sentinel-1 orbital state vectors provided by ESA, …"

Page 10, lines 4-6: It looks like these error estimates assume that the orbital errors apply to only one of the images. I think if you assume a 5 cm error in both images, these velocity errors would double? Since the product used the restituted orbits, I think it would also be more appropriate to frame these sentences in terms of the 10 cm accuracy of those data, even though your investigation shows that the measured errors are very similar for both of them.
It is correct that we should account for the error on both images. Since the values are given as RMS the errors on the displacement should be multiplied by $\sqrt{2}$. We have rewritten the paragraph using the values for the restituted orbits and applied the factor of $\sqrt{2}$:
*"For Sentinel-1 data, absolute orbital errors are on the order of 5 cm RMS when using the precise orbit product available after 21 days (Peter et al., 2017). The restituted orbits typically used in the PROMICE product generation are available shortly after acquisition, and have a nominal accuracy of 10 cm RMS. This corresponds to 8.6 m/yr RMS for a velocity measurement using a 6-day pair and 4.3 m/yr for a 12-day pair."*

Page 13, line 9: I'm a bit confused by the use of 'delays' because the timing aspect hasn't been introduced yet as far as I can tell. Would 'shifts' be appropriate, since it's used in the previous sentence?
Agreed, "shifts" is more appropriate. We have changed the text.

Page 13, line 18: 'affecting the ability to measure ice velocity': can you be more specific here and relate this to the cross-correlation procedure?

We have changed the sentence to:

*"Temporal decorrelation is caused by changes in radar backscatter between acquisitions that reduce the correlation between the image patches which are cross-correlated in the offset-tracking procedure, leading to noisy or even missing measurements."*

Page 13, line 21: 'sub-resolution structure' of the snow/ice?

In this case yes, but more generally, the sub-resolution structure of the imaged scene. We have changed the sentence to:

*"Speckle is a property of radar images, caused by variations in the sub-resolution structure of the imaged scene,"*

Page 13, line 23: I think that 'If the scene is moving…. From the same track' is perhaps unnecessary and it would be sufficient just to say speckle can be used to track ice flow when the ice-flow is spatially uniform over the dimensions of the interrogation areas.

It is important that the track is repeated with sufficient precision, since large spatial baselines can also degrade the coherence, although for Sentinel-1 this is generally not a problem due to the tight orbital tube. We have rephrased to:

*"If the ice-flow is spatially uniform and the sensor track does not deviate excessively for the two acquistions (the latter is generally not a problem for Sentinel-1), the speckle pattern can be tracked between acquisitions"*

Page 13, line 24: perhaps 'steep spatial gradients in ice flow', or similar, rather than just 'rapid ice flow'.

Agree, we have changed "rapid ice flow" to "steep spatial gradients in ice flow".

Page 13, line 25: 'the noise level exceeds the signal' is a bit confusing (and impossible by definition?). Perhaps 'the signal to noise ratio is low' would be sufficient?
Page 13, lines 25-26: 'by averaging multiple measurements to reduce the noise' – can you specify what you mean by measurements in this context? Multiple shift maps in the mosaic? Or spatially over multiple pixels?

We have changed the wording to:

"Often in the interior, the signal-to-noise ratio is low, but since up to five velocity maps from each track are averaged to produce the PROMICE product, the noise can be reduced."

Page 13, line 28: It's not clear what the difference between 'noisy' and 'patchy' is in this context. Would one or the other suffice?

This is correct, from the preceding discussion it is implicit that the measurements are noisy. We have changed the wording to:

*"In extended homogeneous areas of low coherence, the velocity measurements can become patchy, since many unreliable measurements will be discarded by the culling procedures"*

Page 15, line 4: We have used a variational stationary noise filter to good effect to remove ionospheric striping in the velocity estimates. The method we applied to the Sentinel 1 velocity estimates is described very briefly in Tuckett et al. (2019), and the underlying algorithm is described in Fehrenbach et al. (2012). I think the code is documented here: https://www.math.univtoulouse. fr/~weiss/Codes/VSNR/VNSR_VariationalStationaryNoiseRemover.html

Fehrenbach, J., Weiss, P., and Lorenzo, C. 2012. Variational algorithms to remove stationary noise: applications to microscopy imaging. IEEE Transactions on Image Processing. DOI: 10.1109/TIP.2012.2206037.

Tuckett, P. A., Ely, J. C., Sole, A. J., Livingstone, S. J., Davison, B. J., van Wessem, M., Howard, J. 2019. Rapid accelerations of Antarctic Peninsula glaciers driven by surface melt. Nature Communications, 10, 4311. DOI: https://doi.org/10.1038/s41467-019-12039-2.

This is an interesting approach, and something for us to look into, although its performance should be assessed in the typical case where there can be quite a lot of missing data in the individual pairs.

Page 17, line 2: See my comment re Figure 10. Perhaps referring to regions or low/high coverage, rather than regions of blue/yellow in Figure 10 would be clearer?

Yes it makes more sense -we have changed the wording accordingly. See also our response to your comment on Figure 10.

Page 17, line 8: Can you clarify what you mean by 'amount of data'?

Yes, 'amount of data' is changed to 'number of acquisitions'.

Page 19, line 2: 'properties observed by the radar' is a bit vague. Can you be more specific here, for example by referring to coherence or speckle?

Yes, it now reads: '..-both processes leading to loss of coherence.'

Page 19, line 6: It's not clear to me what this means – does it mean high standard deviation in the velocity maps? Or some other measurement of uncertainty?

We have deleted this sentence. It is more confusing than informative.

Page 21, lines 21-22: 'outer most parts of the outlet glaciers still have reasonable coverage' is a little vague. Can you clarify what you mean by 'outer'? And by 'reasonable' do you mean that it is better than the 2-cycle PROMICE product?

Agreed this part is a little vague. What we mean is that when increasing the temporal resolution there is still data (although with increased noise) on the outlet glaciers -it mainly the ablation areas with lower flow speeds that are affected. We have changed the wording -see below. See also our response to your comment below.

"It worth noting that the outer most parts of the outlet glaciers still have reasonable coverage and for studying changes in fast flow in these areas the increased temporal resolution may outweigh the downsides."
->
"For studies concerned with changes in fast flow in these areas the increased temporal resolution may outweigh the downsides."

Page 21, line 22: 'the increased temporal resolution may outweigh the downsides' – have you looked at a product using all 6 and 12 day pairs, but only over 1 cycle/12 days? I guess it will have reduced coverage and perhaps be less smooth, but may be useful for investigating the outlet glaciers.

The 6dOnly_1cycle is the product you are suggesting. It takes 12 days to cover all Greenland twice with a temporal baseline of 6 days, so there are no 12 day pairs that can be included. Figures 9 and 10 and Table 5 show that this product has lower coverage and is more noisy than the 2 S1A-cycle products due to the shorter temporal baseline of the included pairs, the S1A-B geolocation bias and simply because it contains fewer pairs. As you mention, the higher temporal resolution could easily outweigh these 'downsides' in cases, where the user is interested in the fastest flowing parts of the ice sheet. For that reason, the 6dOnly_1cycle may very well be a future product within PROMICE.

Page 24, line 5: This is the first mention of the extra log following the winter campaigns. I wonder if it would be better mentioned earlier?

Agree, we have included a sentence in the Section 'The PROMICE velocity product':
'However, during the winter campaigns where more data is acquired this lag may be larger.'

Page 24, line 18: 'vary' should be 'varies'
Done

Page 24, line 20: I suggest adding ', which hinder velocity retrieval' or similar after 'high precipitation rates'
Done

Page 24, line 25: add 'here' after 'presented'?

Done

**Next, I list some minor spelling and/or grammatical errors and suggestions:**

✓ Page 1, line 2: comma before 'which'

✓ Page 1, line 4:'span' should be 'spans'

✓ Page 1, line 6: I think '6 and 12 day' should be '6- and 12-day' (here and throughout)

✓ Page 1, line 18: 'Greenland Ice Sheet' should be GrIS

✓ Page 2, line 10: 'SAR' should be defined on first use.

✓ Page 2, line 25: 'Greenland wide' should be 'Greenland-wide'

✓ Page 2, line 25: 'IPP' should be defined on first use.

✓ Page 2, line 30: There should be a space after 'i.e.'

✓ Page 3, line 3: 'timeseries' is sometimes given as 'time series'. I'm not actually sure which is correct, but I think you should be consistent.
We will use 'time series'.

✓ Page 3, last line: should 'Wideswath' be 'Wide (IW) swath'?

✓ Page 4, line 10: '5' should be 'Section 5' (here and elsewhere)

Page 9, line 21: should that be four groups instead of 'three'?

✓ Page 10, line 8: I think 'Southwest' should be lowercase (there are similar errors throughout the manuscript).

✓ Page 13, line 8: m/yr instead of m/s

✓ Page 13, line 31: needs a comma after 'Section 4.2'

✓ Page 14, line 4: Does 'Total Electron Content' need to be uppercase?

✓ Page 15, line 16: '(Boncori et al., 2018)' should be 'Boncori et al. (2018)'

✓ Page 15, line 28: 'can actually' – should this be 'could actually'? Or did you test this specifically?

✓ Page 16, line 10: 'in the top panel', should instead just refer to the Figure/panel number/label. (same comment for line 15)

✓ Page 17, line 14: 'surface-properties' doesn't need to be hyphenated.

✓ Page 17, line 20: need a comma after 'baseline'

✓ Page 20, first para: I found this a bit repetitive. I wondered if the penultimate sentence could be merged with the first sentence to streamline it a bit.

Agree. We have also inserted a missing data reference. The first paragraph in that section now reads:

*'We validate the PROMICE ice velocity product against in-situ GPS measurements. Only a limited number of GPS measurements are available since the data should overlap in time with the period of the PROMICE ice velocity product and have a a temporal resolution comparable to or higher than the PROMICE ice velocity product. Furthermore, the measurements are biased toward the slow moving 5 parts of the ice sheet ablation zone. We compare the PROMICE ice velocity product to in-situ GPS data from the PROMICE automatic weather stations (AWS) (van As et al., 2011). Locations are displayed in Fig. 11.'*
->
*'We validate the PROMICE ice velocity product against in-situ GPS measurements from the PROMICE automatic weather stations (AWS) (van As et al., 2011; Fausto and van As, 2019) and perform an analysis over stable ground. Only a limited number of GPS measurements are available since the data must overlap in time with the period of the PROMICE ice velocity product and have a temporal resolution comparable to or higher than the PROMICE ice velocity product. Furthermore, the measurements are biased toward the slow moving parts of the ice sheet ablation zone. Locations are displayed in Fig. 12.'*

✓ Page 20, line 25: suggest adding 'roughly' before 'east/west'

✓ Page 21, line 12: is 'Supp' referring to the supplementary information in Hvidberg et al. (2019)? It's not really clear whether it's a typo.

No, we are also unsure how to cite the supplementary material correctly. We have changed the text to:
*'...values of 1.5 and 1.4 m/yr, respectively (supplementary material in Hvidberg et al., 2020). '*

✓ Page 21, line 21: 'noticing' should be 'noting'? -> this sentence has been changed and no longer includes 'noting'..

✓ Page 21, line 32: 'spatially better' sounds a bit odd to me. Perhaps 'A validation dataset that is not biased…' would suffice? And/or mention that a spatial distribution representative of a greater range of observed ice velocities/flow regimes would help?

Agree, the sentence now reads:

*A spatially better distributed set of validation data, which is not biased towards slow flowing areas in the ablation zone would help assess whether the reported product errors capture this correctly.*
->

*A validation dataset which is not biased towards slow flowing areas in the ablation zone but is representative of a larger range of flow regimes and surface conditions would help assess whether the reported product errors capture this correctly.*

**Below, I provide some feedback on the figures and figure captions**

Figure 2: Are you sure the blue polygons represent radar image footprints? Some of them seem too large. Looking online, it looks like they might be acquisition segments instead? (Though I couldn't find a clear explanation of the difference). Can you also provide the dates for the 12-day periods shown?

The polygons represent the full segments acquired by the radar over Greenland, for each track. For practical reasons, these are split by ESA and delivered as 250 km x 250 km slices, which are designed to be easily concatenated into longer image strips. As for the dates, Fig.2a represents the tracks which are observed on every cycle, whereas the coverage shown on Fig.2b is for a single orbital cycle of the winter campaign 2019-2020 (repeated over 4 cycles). We have changed the figure caption to:

*"Typical Sentinel-1 coverage over Greenland for a single 12-day orbital cycle, (a) during the standard observation scenario, (b) during the dedicated winter campaign from December 2019 - February 2020. The blue polygons represent acquisitions from different tracks, acquired at different times during the cycle."*

Figure 3: Should the culling prior to mosaicking be included in the flow chart?

The flowchart has been updated to include details on the offset-tracking, including the culling prior to mosaicking.

Figure 4, caption: I don't think 'North' should be capitalised.

Figure 5, caption: Needs a space after 'a)'

Yes, a space has been inserted.

Figure 6: axis labels should have Va and Vr as $V_a$ and $V_r$

Fixed

Figure 7, caption: 'Pair' should be lower case.

Fixed

Figure 8, caption: '7' should be 'Figure 7'. 'Pair' should be lower case in both instances.

Fixed

Figure 9: Panels should be labelled a/b. I'm not sure the panel titles add much here either. The legend in the bottom panel blocks some of the data – can the scale of the yaxis be changed so that it doesn't block the lines, or can it be placed outside the graph?

Yes, the panels are now correctly labelled and the legend in the lower panel has been shifted, so it doesn't block the data. We have expanded the figure captions to include a better description of the figure.

Figure 10: I wondered if you could calculate the number of days in an average/given year for which there is velocity data in each pixel? That seems more intuitive to me than the current unit on the colour bar. Using something like that might also make it easier to refer to the values in the text. I think it would also be helpful to label the areas mentioned in the text.

What we aim to show with Figure 10 is where the user can expect good temporal data coverage throughout the time series and how it changes if not all possible pairs are included or if temporal resolution is increased. The temporal coverage depends on: * The number of acquisitions covering the grid point, which again depends on the time of year, the location and the time span of the product (the PROMICE product includes more pairs than any of the three other time series) *How often coherence is lost/ how often the processing fails. I am not sure how to calculate the number of days where there is coverage in a good way, and the figure in its current form shows the issues we want to highlight. We have labelled Figure 10 a) so it shows the locations discussed in the text.

Accounting for the issues discussed above, we have changed the caption so it better describes the figure:

*'Effect of including 6- and 12-day pairs on coverage in the mosaic: a) The PROMICE ice velocity product b) 6dOnly time series c) 12dOnly time series and d) 6dOnly_1cycle time series'*
*->*
*'Temporal coverage: Spatial view of the percentage of all mosaics in that have data in a given grid point: a) The PROMICE ice velocity product b) 6dOnly time series c) 12dOnly time series and d) 6dOnly_1cycle time series. The numbers in a) indicate the locations of the areas mentioned in Sec. 6: Areas where SAR in IW mode has not been acquired on a regular basis: 1, 2 and 3 refer to the triangular area in Melville Bay, North Greenland and the Scoresbysund area, respectively. Areas with low ice velocity coverage: 4, 5, 6 and 7 refer to the southeast ice sheet margin, small area in South Greenland, an area north of Rink Glacier and the Melville Bay area, respectively.'*

We have changed the wording of the paragraph starting on P 17 L1:

*'Figure 10 provides a spatial view of the fraction of all mosaics that have data in each grid point for each of the time series. Blue colors indicate that a grid point rarely has data, while yellow indicates a temporal coverage close to 100%. All four time series have a large blue area in the ice sheet interior, where SAR data in IW mode is rarely acquired as is evident from Fig. 2. The same explanation is true for the smaller triangular areas in the Melville Bay area and northern Greenland as well as the Scoresbysund area. However, the large blue/green area along the southeast ice sheet margin as well as an area in southern Greenland, one north of Rink Glacier inWest Greenland, and one in the Melville Bay area all have routine SAR IW acquisitions every 6 days. This will be discussed in the following.'*
*->*

*Figure 10 provides a spatial view of the temporal coverage of the time series. It shows the percentage of all mosaics that have data in a given grid point for each of the time series. Blue colors indicate that a grid point rarely has data, while yellow indicates a temporal coverage close to 100% . A number of circumstances influence the temporal coverage: The more acquisitions cover a grid point, the more likely it is to have a pair where coherence is not lost. The number of acquisitions depends on the time of year, the location and the time span of the product (the PROMICE product includes more pairs than any of the three other time series) The temporal coverage also depends on how often coherence is lost leading to how often the processing fails.*

*All four time series have a large low-coverage area in the ice-sheet interior, where SAR data in IW mode is rarely acquired as is evident from Fig. 2. The same explanation is true for the smaller triangular areas in the Melville Bay area and northern Greenland as well as the Scoresbysund area (locations 1, 2, and 3 in Fig. 11 a). However, the large area with low coverage along the southeast ice sheet margin as well as an area in southern Greenland, one north of Rink Glacier in West Greenland, and one in the Melville Bay area all have routine SAR IW acquisitions every 6 days (locations 4, 5, 6, and 7 in Fig. 11 a). This will be discussed in the following.*

Figure 12: Inconsistent use of 'IV' and 'ice velocity' in the caption. Axis label font is a bit small. Rather than title the panels, I think labelling them a-c and describing them in the figure caption would be clearer. Axis labels should also be consistent with previous plots and text (i.e. use vx and vy).

'IV'' has been changed to ice velocity both in the caption but also in the figure labels. We have changed the titles of the subplots to better describe what the figure shows, and the subfigures has been labelled a, b and c.The figure caption has also been improved and now reads:

*'Scatterplots of PROMICE GPS IV vs PROMICE ice velocity.'*
->
*'Scatterplots of PROMICE GPS ice velocity vs PROMICE ice velocity. a): Scatter plot of the magnitude of the velocity. b): Scatter plot of the vx-component. c): Scatter plot of the vy-component'*

Table 4, caption: 'info' should be 'information'.
Agree -done.

**Reviewer 3**

Review of Solgaard et al.
Greenland ice velocity maps from the PROMICE project

This manuscript describes in detail an ice velocity (IV) products derived within the framework of the Programme for Monitoring of the Greenland Ice Sheet (PROMICE). IV products generated span an observation period of 24 days, are provided every 12 days and posted about 10 days after the last acquisition in the observation period. They are provided at 500 m posting in Polar Stereographic projection. The authors describe the Sentinel-1 data and ancillary data sets used for product generation, detail the automated product generation process and provide a comprehensive error assessment.

**General comment:**

This is one of several projects generating IV products for the Greenland ice sheet. The authors mention a number of other products in the introduction. What would be interesting here is how PROMICE IV fits within this lot given that the products are assessed in detail with slow moving areas as well as compared against GPS measurements. The authors encourage feedback from the community, so some information for the community through an inter-comparison of products as part of the product assessment would be considered an asset. The comparison with sub-sets of the product is a good first step, but seems insufficient.
The manuscript lacks justification regarding some of the product parameter choices. These parameters seem to be data driven (as opposed to science driven), which is fine, but some more discussion of trade offs would be helpful. Section 6 addresses some of this, but it does not fully explain all decisions.

- What drives the 500 m grid? Is this a suitable choice for all glaciers is Greenland, and if not, what percentage of glaciers is affected?

The choice of 500 m is driven by the spatial resolution of the measurements. Unfortunately this is not fixed for offset-tracking methods and not known a priori, since it is partially data dependent.
A lower bound is provided by the measurement spacing, which is 40 x 10 pixels in slant-range and azimuth, corresponding to 150 x 150 meters on ground. This would be the case for a measurement window containing a radar point target. However, this is not very realistic for ice-sheets, since, when the measurements rely on features, they are typically distributed targets, such as crevasses with a similar orientation. The upper bound on spatial resolution is instead determined by the cross-correlation window size, which is 256 x 64 pixels, corresponding to 800 x 900 meters on ground. 500 m represents a compromise between the lower and upper bounds, tending towards the latter, which is more representative of the true spatial resolution, since we are tracking either distributed targets, close to the ice-sheet margin, or intensity speckle, on the ice-sheet interior.

In the abstract and section 2, we have changed *"spatial resolution"* to *"grid spacing"* and we have added the following to section 2:

*"The effective spatial resolution is on the order of 800-900 m, determined by the fixed size of the correlation windows used in the offset-tracking (see Sect.4.2). Thus, glaciers smaller than approximately 1 km across will not be fully resolved."*

The following has been added to section 4.2:

*"At each grid point, surrounding image patches of 256×64 complex pixels (slant range×azimuth) are extracted in both SLCs.This patch size has been chosen to maximize the coverage over different flow regimes and coherence levels, and means that the product has an effective spatial resolution on the order of 800-900m.As shown in (Boncori et al., 2018), an adaptive window size approach similar to the one described in (Joughin, 2002) could provide for a locally finer spatial resolution when the data allows it, but this is currently not implemented in the IPP processor"*

 - Why the 24 day (two S1A cycles) observation period? What is gained by averaging more (or less) data, what is lost? The discussion on page 19 does not feel sufficient.

We have added a summarising paragraph at the end of Section 6 to make our choice clearer based on the pro and cons of analysis carried out in the section:

*'The analysis from this section shows, that it is possible to provide a Greenland-wide ice velocity product with a higher temporal resolution than the PROMICE product (the 6dOnly_1cycle product), but also that this comes with the price of reduced spatial coverage and higher uncertainty. Creating a product spanning more than two Sentinel-1A cycles will have opposite effects: a reduction in uncertainty, a (small) increase in spatial coverage and reduced temporal resolution. The two Sentinel-1A cycles choice for the PROMICE product is therefore a compromise between having reasonably high temporal resolution and good coverage and reducing noise. '*

Different types of products are appropriate for different purposes: some users require high temporal resolution while others need complete coverage and low uncertainty.

 - What drives the 10 day lag? It should be said that product generation 10 days after the last acquisition is impressive, the question is: Would 11 or 15 days be sufficient, is there anything to be gained for this to be 9 days?

The 10 day lag is simply the time it takes to include the last acquisition in the processing, do post-processing and have a time-buffer. We also state in the manuscript, that the 10 day lag is an aim and that it may take longer (or shorter) before the product is available. Stating when users can expect the product to update as well as it being relatively close to the time of the last acquisition are the most important aspects of this. Having a lag of 15 days vs 9 day most likely matters less, but for monitoring ice discharge and dynamics a lag of several months is not sufficient.

- With Landsat-8 and Sentinel-2 openly available, why the focus on Sentinel-1 for the product? This is not meant as criticism but a request for justification of the choice.

We are interested in obtaining mosaics all year round, and for this SAR data is great. It is not limited by clouds or darkness as is optical data (from e.g. Sentinel-2 or Landsat). Of course, using SAR data has limitations and including optical based ice velocities would enhance the product especially during summer where surface melt is a problem for SAR based products.

Section 3: Precision orbits
The web site provided http://aux.sentinel1.eo.esa.int/POEORB/ is outdated as of early March 2021. ESA has switched to another site (tbc) for the new orbits. Information for the products you refer to are still on the old site (as per review submission). Please provide the new site information as well.

The old website is no longer available - we have provided instead the new link: https://scihub.copernicus.eu/gnss/

Section 4.5: Culling
The existing time series data set is used to provide an average velocity to drive the culling of outliers. Seasonal variation of the ice speed is mentioned as an issue as it can exceed 200%. Would a seasonally limited average make a difference here, given that 4 years of data are already available?

It will certainly be worthwhile to improve the post-processing culling procedure in the future, and it is something we are working on. Seasonality would be a way to go, but the method would also have to not remove signals stemming from surge dynamics and longer term slow down or speed up. The culling method presented here is conservative, and not all outliers will be removed.

Section 5: Error
While not a big issue in Greenland, floating tongues have a tidal driven vertical displacement component that will be interpreted as speed if not corrected. At the very lest, this should be accounted for in the error

This is important to emphasize. We have added a sub-section (5.5) on tidal motion errors with the following text:

*"A few outlet glaciers in Greenland (Petermann and 79 Fjord being the most prominent) are characterized by having a floating tongue subject to tidal motion. The tidal motion introduces a vertical shift, with the sensitivity of this shift to the tidal signal increasing from 0 near the grounding line to 1 on the fully floating part of the tongue, a transition zone which is typically 5-10km wide (Padman et al., 2018). This vertical shift will affect the ice velocity estimate, as the difference in the shift between the two radar acquisitions is projected on to the radar line-of-sight and interpreted as motion in the slant range direction. In (Reeh et al., 2000), tidal-induced shifts of approximately +/-0.5 m were observed using GPS receivers placed on the floating part of 79 Fjord glacier. This could, for a 6-day pair, lead to errors in*

*the ice velocity estimate of more than 50 m/yr, although the averaging of several acquisitions in the PROMICE product will tend to reduce this error. The effect is not modeled in the PROMICE product, so care should be taken when using the product on floating glaciers*

Section 5.4: Ionosphere
The biggest impact of ionosphere perturbations are in azimuth direction. The available data are acquired in a way that for quite few regions you have ascending and descending data acquisitions available. Why not use the range - range components available in those regions to minimize the error?

We have adressed this in the answers to reviewer 1, which we repeat below:

The ionospheric effects are indeed the major error source, and we have experimented with methods relying on the dispersive properties of the ionosphere, similar to the ones described in (Gomba 2018) and (Liao et. al., 2018). However for Sentinel-1 TOPS data, there is a problem with block processing artifacts present in most Sentinel-1 SLCs, which severely hampers this approach. Another method to reduce the impact of ionospheric effects is to exploit the fact that in some regions, measurements from both ascending and descending tracks are available, and in this case one can derive the horizontal velocity from only the range offsets, which are much less affected by the ionosphere. This is being worked on, and will be included in a future update of the processor. We have added the following text:

*"Another method to reduce the impact of ionospheric effects is to exploit the fact that in some regions, measurements from both ascending and descending tracks are available, and in this case, ice velocities can be derived from only the range offsets -- which are much less sensitive to ionospheric effects -- and the SPF assumption(see Sect.4.3). In the standard S1 acquisition plan (Fig. 2a), only two ascending tracks are acquired (the long track along the west coast of Greenland, and the track covering the northeast margin of the ice sheet up to the northernmost point), so the method will not be applicable everywhere. During the winter campaigns ( (Fig. 2b), this method will be applicable in a much larger part of the ice sheet. Work is undergoing to include this method in a future update of the PROMICE product."*

Section 7: Validation
The detailed validation against GPS measurements is appreciated.
Here, the evaluation of other existing Greenland IV products would have been a useful add On.

We agree that an evaluation of the various ice velocity products that are available is interesting, but we also find that this is out of scope for this data paper. In the paper, we validate our product against in-situ measurements, include an analysis over stable ground/ ice free areas as well as include the study by Hvidberg et al, 2020. All of which show that product performs well. We agree that it would be preferable to include in-situ measurements of fast flow, but we have not been able to find any. Comparison to other products would not solve this issue, since they have not been ground-truthed either at higher speeds.

Are the GPS data also available as a product? If so, please provide access information.

Yes, thank you. The data reference has been added.

 Section 9: Summary and Outlook
Please provide the initial motivation for the product upfront

Page 24, lines 25,26:
" The PROMICE ice velocity product presented was originally intended primarily to
calculate ice discharge through marine terminating glaciers of the GrIS as done in Mankoff
et al. (2020)."

Yes, we already describe some of the uses of the product in the Introduction. We have expanded this
sentence to:

*"The product is used as input to, for example, the solid ice discharge product by Mankoff et al. (2020) and
to study GrIS wide glacier dynamics in high temporal detail in Vijay et al. (2019)."*
->
*"The product is used as input to, for example, the solid ice discharge product by Mankoff et al. (2020) on
a routine basis  and to study GrIS wide glacier dynamics in high temporal detail in Vijay et al. (2019)."*

**Figures:**
General comment: Sub-figures are not consistently named

 Figure 2:
Based on the coverage maps the products have regionally different number of IV
estimates and this number varies by season (with more Tracks covering the ice sheet
during Winter). This is mentioned in Section 6 (page 19). Are the numbers for the
estimates on a per pixel basis provided in the product somewhere? This seems relevant
when comparing different maps.
This information is not provided in the PROMICE product, instead it is intended that the standard
deviation maps be used for this purpose. Through the weighted averaging and standard deviation
calculation, the fusion described in section 4.4 takes into account both the number of measurements
and the quality of the measurements for each pixel. We believe this information is easier for the end
user to apply than the number of measurements at each pixel.

 Figure 3:
With the observation period set to 24 days, the minimum and maximum pair numbers are
known and could be reflected in the figure.
The figure is meant as a high-level overview of the processing, and the number of pairs is the total
number of data pairs (on all tracks) processed for the two S1A cycles comprising the PROMICE product. It
will therefore vary, e.g. N would be much higher during the winter campaign. The figure has been
updated to include offset-tracking details.

 Figure 4:

Figure 4 and the corresponding discussion on page 9 would benefit from an assessment how many data points are culled (vs. how many outliers are not culled) for the various parameter selections.

Agree. We have included this information on P9 L10:
"For $k_{thr}$=3, the (real) summer speed up near the front is conserved, while the majority of spikes further inland are removed. Applying a stricter value, $k_{thr}$=1 removes not only outliers but also the real signal due to summer speed-up."
->
"For $k_{thr}$=3, the (real) summer speed up near the front is conserved, while the majority of spikes further inland are removed. Applying a stricter value, $k_{thr}$=1 removes not only outliers but also the real signal due to summer speed-up. In this case, 8 % of the pixels are culled while 4 % of the pixels are culled when applying $k_{thr}$=3."

Figure 5:
Figure 5 would benefit from a couple of insets providing more spatial detail of the culling.

We have expanded the figure to include zoom-ins of areas in western and eastern Greenland.

Fitures7, 8:
Figures 7 and 8 would benefit from being placed on the same page (or they should be combined to a single figure)
The figures have been placed together.

/ Figure 9:
Figure 9 has sub plots, should they not be a) and b)? Also, sub-figure annotation is inconsistent between figures.

Yes, a and b have been added to the subplots.

Product coverage is shown in multiple figures (1,2,5,(7,8),9,10 ), most of which provide spatial information. There is no such spatial information for the errors characterized by the STD shown in Figure 9. It would be useful to add the errors to one of the figures showing example maps (maybe Fig 1).

We have added a figure at the end of Section 5.5 Error Estimation showing the error estimate for the same mosaics we show in Figure 1. We have added a line of text at the end of the paragraph starting P15 L12:
'A Greenland-wide view of the error estimate for the PROMICE product is given in Fig. 8 for the same mosaics displayed in Fig. 9.'

Figure 9 product coverage indicates that culling is depending on the season. This

seasonality is of interest and could be shown in more detail.The authors use the STD as a proxy to estimate the error of the product. Does this hold in
the presence of strong ionospheric perturbations? In such cases the worst streaks are
culled but still have large area offsets causing a higher error in the product.

More points are culled in summer than in winter. We have added a time series to Figure 9 showing the percentage of points that are culled for each mosaic.  Regarding ionopsheric perturbations, the impact of these are not fully reflected in the standard deviation estimate, due to the spatial correlation. We believe this is already described in Section 5.6 (Error estimates):

[revised manuscript text omitted]

---

## Author Response (AR2)

Dear Editor,

This is excellent news! We have made the following changes to the manuscript following your comments:

1. P19, L19: "Figure 11 provides a spatial view of the …" should be changed to "Figure 10 provides a spatial view of the …"

*The figure number is correct, but we agree that the wording "temporal coverage" sound like Figure 10. We have removed 'temporal' from the text in order to avoid confusion.*

*Figure 11 provides a spatial view of the temporal coverage of the time series. -> Figure 11 provides a spatial view of the coverage of the time series.*

2. P2, L29: "IPP" should be defined as "IPP (Interferometric Post Processing)" because it's the first time an abbreviation has appeared.

*Done.*

3. P10, L28: "three main groups" should be changed to "five main groups".

*Done.*

Thank you for acting as editor on our manuscript.

SIncerely, Anne Solgaard on behalf of all co-authors